# RBIO1 - TRAINING SCIENTIFIC REASONING LLMS WITH BIOLOGICAL WORLD MODELS AS SOFT VERIFIERS

## ABSTRACT

Reasoning models are typically trained against verification mechanisms in formally specified systems such as code or symbolic math. In open domains like biology, however, we lack exact rules to enable large-scale formal verification and instead often rely on lab experiments to test predictions. Such experiments are slow, costly, and cannot scale with computation. In this work, we show that world models of biology or other prior knowledge can serve as approximate oracles for *soft verification*, allowing reasoning systems to be trained without additional experimental data. We present two paradigms of training models with approximate verifiers: **RLEMF**: reinforcement learning with experimental model feedback and **RLPK**: reinforcement learning from prior knowledge. Using these paradigms, we introduce **rbio1**, a reasoning model for biology post-trained from a pretrained LLM with reinforcement learning, using learned biological models for verification during training. We demonstrate that soft verification can distill biological world models into **rbio1**, enabling it to achieve state-of-the-art performance on perturbation prediction in the PERTURBQA benchmark. We present **rbio1** as a proof of concept that predictions from biological models can train powerful reasoning systems using simulations rather than experimental data, offering a new paradigm for model training.

## 1 INTRODUCTION

Building foundation models suitable for scientific tasks is a task of major interest and has produced numerous successful examples in recent memory (Abramson et al., 2024; Cui et al., 2024; Lin et al., 2023). Similarly, large language models (LLMs) have shown groundbreaking potential as parametric representations of the world's knowledge, and have been used across every sector. A key challenge is figuring out how to bridge the quantitative accuracy of models of experimental scientific data, for example in biology, with LLMs such that knowledge from these low-level representations of biological systems may be transferred into more flexible and interactive models, such as conversational LLMs, with the explicit goal of being useful for scientific exploration.

Of great promise on scientific tasks are reasoning models, which aim to extend LLMs toward systems that can perform structured, multi-step inference and use test-time compute to generalize better to a given query. Popular reasoning models like DeepSeek-R1 (Guo et al., 2025) and QWEN (Team, 2024) have shown potential in multiple fields, while specialized reasoning LLMs have been explored in fields such as medicine (Fallahpour et al., 2025; Cao et al., 2025) and chemistry (Narayanan et al., 2025). In frameworks such as reinforcement learning with human feedback (RLHF) (Christiano et al., 2017; Stiennon et al., 2020), and reinforcement learning with verifiable rewards (RLVR) (Pan et al., 2023), both experimental data collection with human labels and exact oracles of rewards are used to train language models to align to a reward structure and improve their reasoning capabilities. In domains that are not formally specified like biology, however, experimental data and ground-truth verifiers are scarce: while mathematics and code benefit from exact execution and have symbolically accessible oracles, experiments are costly and slow. Consider training a language model to answer biological queries like '*Will knocking down gene AARS in liver cells affect the expression of gene ATAD2B?*' In traditional RL domains, we could automatically verify thousands of such predictions, but in biology, each verification requires a costly laboratory experiment, making it impossible to

generate the millions of training signals needed for effective learning. This motivates exploring alternative supervision strategies for reasoning for such domains.

To overcome these limitations and further advance the utility of reasoning models for scientific tasks in biology, we propose employing models of biological data to run virtual experiments which can be used as sources of probabilistic -or soft- verification signal. This can be seen as a form of reinforcement learning from AI feedback (RLAIF) (Lee et al., 2023) with structural adjustments to map to our scientific setting, where RLHF and RLVR are not tractable. We consider those *soft verifiers*, since they return probabilistic rewards which measure the coherence of a biology-model or of biological prior knowledge to a reasoning trace and its returned answer. Much like with RLVR, we can use this *soft verification* paradigm to generate a broad distribution of verified data limited only by how we can query the biology model at hand. We thus turn a (world) model of biology into a reasoning environment to generate rewards to train reasoning models.

Our work also connects with the concept of virtual cell models (VCMs) (Bunne et al., 2024; Slepchenko et al., 2003; Loew & Schaff, 2001), which envisions building powerful predictive systems of biology that can simulate transitions such as diseased $\rightarrow$ healthy states. Advances in compute and large-scale data have enabled construction of such foundation models in specific modalities-transcriptomics (Rosen et al., 2023; Pearce et al., 2025; Bian et al., 2024; Ho et al., 2024; Theodoris et al., 2023), imaging (Gupta et al., 2024), proteomics (Abramson et al., 2024; Lin et al., 2023), genomics (Nguyen et al., 2024), and multimodal models (Rizvi et al., 2025; Richard et al., 2024; Levine et al., 2024; Schaefer et al., 2024; Choi et al., 2024; Istrate et al., 2024).

Our approach can be seen as using and aligning such world models of biology into a common representation using language as the bridge. This approach not only aggregates knowledge but also makes it accessible through natural language, allowing experimentalists to interact conversationally with biological models. By distilling biological knowledge into LLMs, we transform experimental insights into human-readable reasoning models. Our motivations are threefold: (i) enable training from biological simulations rather than costly experimental data, (ii) integrate diverse models of biology into a single platform, (iii) democratize access to biological knowledge through dialogue.

**Contributions.** Our work contributes to the design of supervision strategies for reasoning LLMs for scientific use, using biological perturbation prediction -e.g., predicting effects of gene knockdowns on differential expression, as a case study:

1. We propose two new processes for training models with AI-verifiers: **RLEMF**: reinforcement learning with experimental model feedback and **RLPK**: reinforcement learning from prior knowledge - that reward with predictive models, and prior knowledge, respectively.

2. RLEMF-trained models generalize OOD and compete with ablation-models trained on experimental data, achieving new state-of-the-art results on the PerturbQA benchmark

3. We show that mixtures of AI-verifiers can be combined to compose stronger models while drawing from different sources of biological knowledge, even when supervision is off-task.

4. We show that inference-time chain-of-thought prompting (Kojima et al., 2022) further improves reasoning performance, allowing **rbio1** to reach state of the art on the PERTURBQA benchmark without tool use or experimental data at inference, even at a fraction of training data.

In summary, **rbio1** extends standard RL training for reasoning models by incorporating AI-based verification through both predictive biological models of experimental data and curated knowledge sources and provides a general framework of using model simulations to train reasoning models. Code implementing the core training methodology is available at `https://anonymous.4open.science/r/rbio-9155/README.md`. This release focuses on the essential components for reproducibility and community adoption.

## 2 RELATED WORK

Recent reasoning-oriented LLMs-such as OpenAI's o-series, Claude 3.7/4, Gemini 2.5, and DeepSeek-R1-exhibit strong multi-step inference and logical deduction across domains. Their

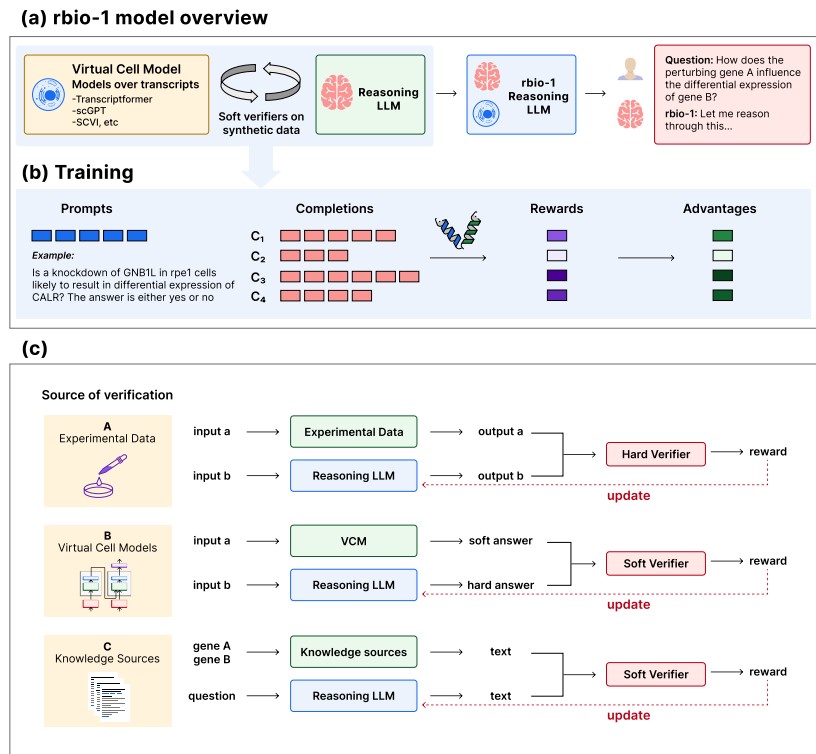

**Figure 1: rbio1 overview.** (a) Distilling VCMs into reasoning LLMs via soft verification. (b) GRPO loop with Virtual Cell Models (VCM) rewards (shown as double helix). (c) Soft vs. hard supervision.

development spans four paradigms: (i) inference-time scaling (e.g., chain-of-thought, self-consistency), e.g., see (Muennighoff et al., 2025); (ii) pure RL approaches like DeepSeek-R1-Zero, where traces emerge from accuracy- and format-based rewards; (iii) hybrid supervised finetuning plus RL, as in DeepSeek-R1; and (iv) distillation into smaller backbones such as Qwen (Team, 2024; Yang et al., 2025) or Llama (Guo et al., 2025; Touvron et al., 2023). Despite advances, persistent challenges remain in hallucination, logical consistency, verbosity, and interpretability-issues directly tied to the quality of the rewards.

Domain-specific reasoning has also been explored. BioReason (Fallahpour et al., 2025) combines a genomic encoder with an LLM for disease-pathway inference with interpretable steps, while Cell-Reasoner (Cao et al., 2025) frames cell-type annotation explicitly as a reasoning task. Both approaches, however, depend heavily on curated datasets, limiting robustness to noisy or rare populations and motivating richer, more scalable reasoning signals. Our approach differs by using machine learning models of biology directly as reward-generating verifiers. Prior methods integrated external models (e.g., embeddings) into reasoning traces but still evaluated against annotated data. We instead shape rewards themselves with model predictions, showing that biological world models can be distilled into reasoning LLMs -positioning our work within the broader space using AI-rewards.

Wu et al. (2025) propose SUMMER, an inference-time pipeline combining knowledge-graph summaries, retrieval, and chain-of-thought prompting for perturbation prediction. While it outperforms prior methods on PerturbQA, gains are modest, causal directionality remains error-prone, and large models are required even for preprocessing. Unlike SUMMER, our models achieve comparable or better results without experimental data, relying solely on model predictions.

Our work also connects to concurrent research on soft- and AI- verification. In RLAIF (Lee et al., 2023) and follow-up work, other LLMs are used as reward mechanisms. Our approach RLEMF 3.3 differs by not requiring an LLM or any text model as an AI-feedback model, and uses models in a different data space of experimental data linked by appropriate prompting techniques and embeddings. Our idea thus builds a bridge between models of experimental data yielding AI-feedback,

**Table 1:** Verifiers used during RL training. EXP = experimental data; MLP = multilayer perceptron; GO = Gene Ontology.

| Verifier | Type | Reward Signal | Source |
|----------|------|---------------|--------|
| EXP | Hard | Binary $r_i^{hard} \in \{0, 1\}$ | Experimental data |
| MLP | Soft | Probability $r_i^{soft} = p,\ 0 \leq p \leq 1$ | Simulations |
| GO | Soft | ROUGE, keyword, likelihood | Knowledge base |

and the reasoning LLMs learning from that feedback to generate more accurate textual descriptions of valid scientific knowledge. However, we share the approach the model is used to provide a probabilistic verifiable reward. Saad-Falcon et al. (2025) also use LLMs as soft verifiers for other LLMs and combine verifiers. In contrast, we generalize beyond LLMs to arbitrary biological models and combine multiple verifiers as separate reward functions. In a framework closest to our approach RLPK 3.4, Yu et al. (2025) use LLMs to use the reasoning LLM itself to score answers as rewards. In RLPK we do not use answers, but structured databases of prior scientific knowledge.

To our knowledge, we are the first to apply this paradigm to reasoning models for biology, shifting the training signal from experimental data to simulations and broadening the design space of verifiers for reasoning LLMs.

## 3 RBIO1: METHODS

In standard domains, during RL training, verifiers return precise signals-for example, whether code executes or a math solution is correct. In biology, some queries can be validated experimentally (hard verification), but exhaustive lab testing is infeasible due to scale. Consider a biological query related to genetic perturbation, such as: *Is a knockdown of AARS in hepg2 cells likely to result in differential expression of ATAD2B?* with a binary answer: yes/no. During training, the LLM produces completions $o_i$ for query $q$. Rewards can be assigned in three ways that we introduce in the following sections and also showcase in Fig. 1. Table 1 summarizes these verifiers and reward formulations. We follow the PerturbQA benchmark protocol (Wu et al., 2025), evaluating CRISPRi single-gene perturbation prediction across four cell lines (RPE1, K562, HEPG2, JURKAT); cell lines share 40–75% of perturbed genes, ensuring out-of-distribution generalization rather than within-cell-line interpolation. A more detailed description of the biological setup is provided in **Appendix A.1**. We report F1, Balanced Accuracy, and MCC as the most biologically meaningful metrics, since identifying true positive perturbations is more critical than avoiding false positives **Appendix A.2**.

### 3.1 REINFORCEMENT LEARNING FOR REASONING

Let $P(Q)$ denote a dataset used for training; $q$ a query sampled from $P(Q)$, $G$ a set of outputs generated during training by the reasoning LLM $\pi_\theta$ in response to input queries; $o_i$ a generated sequence of tokens with tokens $o_{i,t}$ in response to $q$; $\pi_{\text{ref}}$ a reference base model from the supervised finetuned LLM; $r_\phi$ a reward model emitting rewards $r_i$; $L_{GRPO}(\theta)$ the surrogate objective and $\beta$ the coefficient for the KL penalty. Given these variables, Group Relative Policy Optimization (GRPO) (Shao et al., 2024; Mroueh, 2025) training maximizes the following objective function, with the goal of increasing the accumulated collective rewards $\{r_{i,\geq t}\}$:

$$J_{GRPO} = \mathbb{E}_{q \sim P(Q),\ \{o_i\}_{i=1}^{G} \sim \pi_{\theta_{old}}} \left[ L_{GRPO}(\theta) \right]. \tag{1}$$

We use the clipped surrogate objective:

$$L_{GRPO}(\theta) = \frac{1}{|G|} \sum_{i=1}^{G} \frac{1}{|o_i|} \sum_{t=1}^{|o_i|} \min\left( \frac{\pi_\theta(o_{i,t}|q,o_{i<t})}{\pi_{\theta_{old}}(o_{i,t}|q,o_{i<t})} \hat{A}_{i,t},\ g(\epsilon, \hat{A}_{i,t}) \right) - \beta D_{KL}[\pi_\theta || \pi_{\text{ref}}] \tag{2}$$

$$g(\epsilon, \hat{A}_{i,t}) = \text{clip}\left( \frac{\pi_\theta(o_{i,t}|q,o_{i<t})}{\pi_{\theta_{old}}(o_{i,t}|q,o_{i<t})},\ 1 - \epsilon,\ 1 + \epsilon \right) \hat{A}_{i,t} \tag{3}$$

$$\hat{A}_{i,t} = \frac{r_i - \text{mean}(\{r_1,...,r_G\})}{\text{std}(\{r_1,...,r_G\})} \tag{4}$$

$$D_{KL}[\pi_\theta || \pi_{\text{ref}}] = \frac{\pi_{\text{ref}}(o_{i,t}|q,o_{i<t})}{\pi_\theta(o_{i,t}|q,o_{i<t})} - \log \frac{\pi_{\text{ref}}(o_{i,t}|q,o_{i<t})}{\pi_\theta(o_{i,t}|q,o_{i<t})} - 1. \tag{5}$$

## 3.2 RBIO-EXP: Reinforcement Learning with Hard Verification

In this setting, verification relies on experimentally validated observations that provide binary outcomes. Let $D_{\text{exp}}$ denote the collection of experimental results containing pairs $(q, y^*)$, where each query $q$ (e.g., a perturbation experiment) has a corresponding ground-truth label $y^* \in \{0, 1\}$.

We define a *verifier function*

$$V_{\text{exp}}(q, o_i; D_{\text{exp}}) : (q, o_i) \mapsto \{0, 1\}, \tag{6}$$

which returns 1 when the model output $o_i$ matches the experimentally validated outcome for $q$, and 0 otherwise:

$$V_{\text{exp}}(q, o_i; D_{\text{exp}}) = \begin{cases} 1, & o_i = y^*(q) \text{ and } (q, y^*(q)) \in D_{\text{exp}}, \\ 0, & \text{otherwise.} \end{cases} \tag{7}$$

The reward assigned to completion $o_i$ is then defined as

$$r_i^{\text{hard}}(q, o_i; D_{\text{exp}}) = V_{\text{exp}}(q, o_i; D_{\text{exp}}). \tag{8}$$

Here, $D_{\text{exp}}$ is not itself the verifier but rather the data source queried by the deterministic verification function $V_{\text{exp}}$. A description of the RBIO-EXP algorithm is provided in **Appendix Alg- 1**.

## 3.3 RBIO-RLEMF: Reinforcement Learning with Experimental Model Feedback

In many cases, exhaustive experimental datasets $D_{\text{exp}}$ are not available for all biological queries, or are expensive or even impossible to generate. To extend reward coverage across more scientific breadth, we use predictive models of experimental data as *surrogate verifiers*, and denote the process as experimental model feedback. These models -for example, neural predictors of perturbation effects-provide *soft*, probabilistic rewards rather than binary outcomes. More generally, this approach is akin to RLAIF, with the key difference that RLEMF utilizes arbitrary other (non-LLM) models as feedback mechanisms for a query, in our example world models of biology defined on experimental data.

Let $\mathcal{M}$ denote a biological model that can be queried with a prompt $q$ and contextual information $c_j$ (e.g., the cell line or other covariates), producing a scalar prediction $\hat{p} = p(c_j|q; \mathcal{M})$ that reflects the likelihood of the queried biological effect being true.

We define a corresponding *verifier function*

$$V_{\text{EMF}}(q, o_i; \mathcal{M}, c_j) : (q, o_i) \mapsto [0, 1], \tag{9}$$

which emits a soft reward based on the model's predicted probability:

$$V_{\text{EMF}}(q, o_i; \mathcal{M}, c_j) = \mathcal{M}(q; c_j), \tag{10}$$

where higher values indicate stronger agreement between the model prediction and the reasoning output $o_i$. For instance, when $\mathcal{M}$ is a multilayer perceptron (MLP) trained on perturbation outcomes, $V_{\text{EMF}}(q, o_i; \mathcal{M}, c_j)$ corresponds to the model's predicted probability that the statement in $q$ is true for context $c_j$. Within this framework one can also utilize other metrics like log-likelihoods appropriately normalized given a collection of data to be between 0 and 1 cast as rewards representing the belief of the model in the simulated data. The reinforcement learning reward is then defined as:

$$r_i^{\text{soft}}(q, o_i) = \mathcal{M}(q, c_j), \tag{11}$$

which generalizes the hard verification in Eq. 8 to a continuous reward signal in $[0, 1]$.

This approach allows reasoning models to be trained against *world models of biology* that approximate experimental verification, effectively replacing slow or expensive wet-lab experiments with computational feedback. The verifier $V_{\text{EMF}}$ is therefore a deterministic function parameterized by the predictive model $\mathcal{M}$, not a probabilistic conditional, ensuring conceptual consistency with the formulation in Sec. 7. A description of the RBIO-RLEMF algorithm is provided in **Appendix Alg-2**.

## 3.4 RBIO-RLPK: Reinforcement Learning from Prior Knowledge

In addition to experimental or model-based verification, reasoning models can also be guided by structured scientific knowledge. Here, we propose prior knowledge feedback (**RLPK**), where prior knowledge sources are used to verify the semantic consistency of a model's output with curated facts rather than empirical measurements. Let $K_S$ denote a structured knowledge source (e.g., the Gene Ontology) containing a set of prior facts or annotations relevant to a query $q$ which take the shape of text or other sequences. Given an output $o_i$ produced by the reasoning model, we define a corresponding *verifier function* $V_{\text{PK}}(q, o_i; K_S) : (q, o_i) \mapsto \mathbb{R}$, which emits a scalar reward $r_i^{\text{soft}}$ reflecting the agreement between $o_i$ and relevant knowledge retrieved from $K_S$.

**Knowledge retrieval.** For each query $q$, we obtain a collection of knowledge statements $\mathcal{Q}_{\text{prior}}(q) = \{q_j^{\text{prior}}\}_{j=1}^J$ by querying $K_S$ for entries semantically related to $q$ (e.g., gene annotations or biological process descriptions). Each $q_j^{\text{prior}}$ has a sequence length $T_j$ and thus consists of a list of tokens $\mathbf{y}_j = \{y_{j,1}, ... y_{j,T_j}\}$.

**Scoring metrics.** The verifier $V_{\text{PK}}$ computes rewards using one or more of the following metrics:

1. **ROUGE-based score:** We request the model to expose the relevant gene facts inside `<gene_info>` tags—which we refer to as $o_i^{\text{relevant}}$—and compute standard ROUGE-1/2/L F-scores between $q_j^{\text{prior}}$ and the extracted $o_i^{\text{relevant}}$:

$$V_{\text{PK}}^{(\text{ROUGE})}(q, o_i; K_S) = \sum_{j=1}^J \sum_{X \in \{1,2,L\}} \text{ROUGE--X}\big(q_j^{\text{prior}}, o_i^{\text{relevant}}\big),$$

where $o_i^{\text{relevant}}$ denotes the portion of the reasoning trace encapsulated in `<gene_info>` tags.

2. **Keyword-overlap score:**

$$V_{\text{PK}}^{(\text{KWS})}(q, o_i; K_S) = \sum_{j=1}^J \frac{|\,\text{KW}(q_j^{\text{prior}}) \cap \text{KW}(o_i^{\text{relevant}})\,|}{|\,\text{KW}(q_j^{\text{prior}})\,|},$$

where $\text{KW}(\cdot)$ extracts normalized keyword sets from each string.

3. **Likelihood-based score:** For likelihood-based verifiers, we encourage the model to assign higher likelihood to prior knowledge $\{q_j^{\text{prior}}\}$ under our learned policy $\pi_\theta$ given the model's reasoning emissions. To account for variability in sequence length $T_j$, we average over the sequence tokens $y_k$ in $q_j^{\text{prior}}$ and define:

$$V_{\text{PK}}^{(\text{LL})}(q, o_i; K_S) = \sum_{j=1}^J \frac{1}{T_j} \sum_{k=1}^{T_j} \log p_{\pi_\theta}(y_{j,k} \mid y_{j,<k}, o_i, q), \tag{12}$$

which measures how likely each prior-knowledge sequence $q_j^{\text{prior}}$ is under the current policy $\pi_\theta$.

**Reward normalization.** Because raw scores vary widely across metrics, each reward is normalized via an exponential moving average (EMA) before computing GRPO advantages:

$$\tilde{r} \leftarrow (1-\alpha)\tilde{r} + \alpha r_m, \quad \tilde{v} \leftarrow (1-\alpha)\tilde{v} + \alpha(r_m - \tilde{r})^2, \quad \bar{r} = 0.5 + \tfrac{1}{2z_{\max}} \text{clip}\left(\frac{r_m - \tilde{r}}{\sqrt{\tilde{v} + \varepsilon}}, -z_{\max}, z_{\max}\right),$$

where $\bar{r}$ is the normalized reward used for token-level advantages (Eq. 4).

**Summary.** The final reward for RBIO-RLPK is

$$r_i^{\text{soft}}(q, o_i) = V_{\text{PK}}(q, o_i; K_S),$$

where $V_{\text{PK}}$ may be a weighted combination of the metrics above. This keeps consistency with earlier sections: the verifier is a deterministic mapping parameterized by the knowledge source $K_S$, not a probabilistic conditional (see **Appendix Alg. 3**).

### 3.5 COMPOSABLE VERIFICATION FOR MODEL INTEGRATION

All **rbio** models use formatting ($r_{\text{format}}$) and mention ($r_{\text{mention}}$) rewards (e.g., gene mentions). When training with multiple verifiers (Sec. 4.2), each prompt $q$ is verified by a specific function $V_s$. With verifier functions $V_k$ emitting rewards $r_{i,k}$:

$$r_i(q, o_i) = r_{\text{format}} + r_{\text{mention}} + \sum_k \delta_{ks} \lambda_k \, r_{i,k}^{hard/soft}(q, o_i), \qquad \lambda_k \geq 0. \tag{13}$$

Unless stated otherwise, $\lambda_k = 2$, giving soft-verifier rewards higher variance weighting than $r_{\text{format}}$ and $r_{\text{mention}}$ in GRPO updates.

**Use of LLMs**. We used GPT-based tools for minor writing polish and for code assistance in generating plots; all scientific contributions are solely by the authors.

## 4 EXPERIMENTS

### 4.1 RBIO WITH AI-VERIFICATION GENERALIZES OOD ON PERTURBATION TASKS

On PERTURBQA (Wu et al., 2025) (CRISPRi knockdowns in RPE1, K562, HEPG2, JURKAT), models trained with *soft verifiers* generalize to held-out cell lines, reducing reliance on cell-line–specific experimental data. We first evaluate a 2-layer MLP (64 hidden units) trained on three cell lines and use it to generate predictions on the fourth, which serve as rewards during RL. Gene representations include one-hot, Gene2Vec (Du et al., 2019), and ESM (Lin et al., 2023). The resulting models, *rbio-MLP-leave-one-out-one-hot* and *rbio-MLP-leave-one-out-gene2vec*, perform comparably to experimental-data–trained **rbio1** models. The MLP architecture and training details are provided in **Appendix A.6.1**. Supplementary Fig. 13 directly compares the MLP verifiers with the *rbio-MLP-leave-one-out-X* for X $\in$ {one-hot, gene2vec} variations. While the MLP provides calibrated biological supervision, **rbio1** consistently exceeds its performance across all metrics, demonstrating that GRPO-based reinforcement learning refines and extends the verifier's signal rather than merely imitating it.

We compare to two experimental-data baselines: *rbio-EXP-one-cell-line* (train/test within a cell line; Fig. 2a) and *rbio-EXP-leave-one-out* (train on three cell lines; test on the fourth; Fig. 2b). We also benchmark against SUMMER (Wu et al., 2025). As shown in Fig. 2d–e, the soft-verifier models closely match experimental-data models on F1 and MCC, and exceed them in Balanced Accuracy via higher TPR while maintaining similar TNR. Identifying true effects is paramount in perturbation, so higher TPR is valuable even with some F1 trade-off. All **rbio1** variants also outperform GEARS (Roohani et al., 2022), a state-of-the-art model for perturbation prediction and the base Qwen2.5-3B.

In **Appendix A.4, A.4.2, and A.4.3**, we present detailed analyses looking at robustness of **rbio1** models to verifier fidelity and signal miscalibrations, as well as the effect of different reward components. Our results show that **rbio1** models capture genuine biological signal—remaining robust to verifier noise and miscalibration, leveraging but not depending on soft verifier confidence. Beyond perturbation OOD generalization—where shifts reflect cell-line–specific transcriptional programs—we further evaluated **rbio1** on zero-shot transcriptomic cell-state prediction: Alzheimer's disease (2 classes) and myeloid cancers (7 classes). Trained solely on perturbation reasoning with biological soft-verifier rewards, **rbio1** substantially outperforms the base Qwen model on both datasets (+94% F1, +136% Recall for Alzheimer; +36% F1, +68% Recall for Cancer), approaching the performance of SCVI (Lopez et al., 2018)—a model trained on full raw-count matrices—despite using only natural-language inputs over the top-100 expressed genes and metadata. This shows that reinforcement learning with biological verifiers yields transferable representations of cellular state that generalize from causal perturbation dynamics to disease-level transcriptional inference. Full analyses appear in **Appendix A.7**.

### 4.2 TRAINING RBIO1 ON MIXTURES OF AI-VERIFIERS LEADS TO PERFORMANCE GAINS

We find that combining verifiers improves performance over using them individually. Notably, the order in which models see the verifiers matters, reflecting differences in the knowledge provided. For a pair of verifiers $V_i, V_j$, we evaluate:

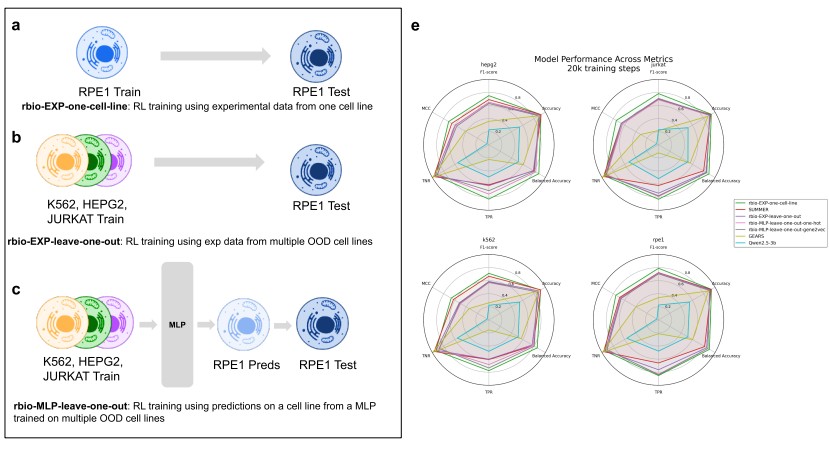

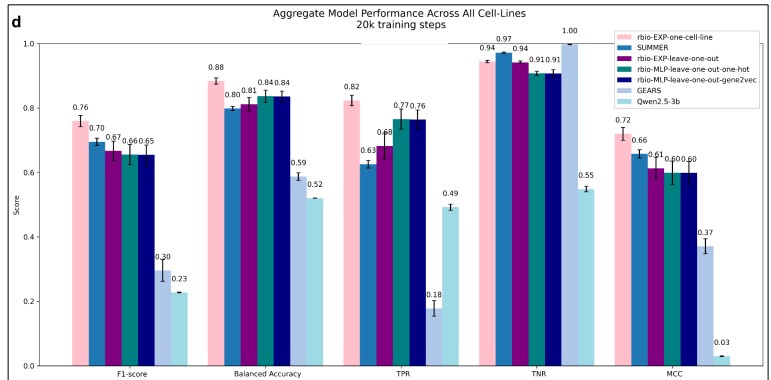

**Figure 2: Model performance for experimental vs. simulation-based soft verification.** **(a)** rbio-EXP-one-cell-line: trained and tested on the same cell line (in-distribution). **(b)** rbio-EXP-leave-one-out: trained on three cell lines, tested on the held-out one (out-of-distribution). **(c)** rbio-MLP-leave-one-out: trained using MLP predictions on the held-out line (MLP fit on the others). **(d)** Aggregate metrics: computed over four cell lines (K562, RPE1, JURKAT, HEPG2), averaged across 5 runs. SEM computation described in **Appendix A.2**. **(e)** Metrics split by cell line. Baselines: *SUMMER* (experimental + domain knowledge), *GEARS* (specialized perturbation model), *Qwen2.5-3b* (base reasoning model).

1. $V_i$: trained only with $V_i$ , $i \in \{1, 2\}$

2. $V_i \| V_j$: trained sequentially, $V_i$, then $V_j$

3. $V_i \cup V_j$: trained on a random mixture of $V_i$ and $V_j$

We experiment with the following combinations of verifiers: (1) $V_1$ = EXP (hard verifier; experimental data) and $V_2$ = MLP (soft verifier; MLP predictions); (2) $V_1$ = EXP and $V_2$ = $GO_{all-ll}$ (soft verifier; GO Ontology likelihood reward); (3) $V_1$ = MLP and $V_2$ = $GO_{all-ll}$. Note that the training data for each of $V_1$, $V_2$ is independent of each other - i.e. if $V_1$ is a verifier of experimental data from a dataset $D_1$, emissions from $V_2$ will be on an independent dataset $D_2$ where $D_1 \cap D_2 = \varnothing$. In the case of the $GO_{all-ll}$ the soft verification is the likelihood of the prior knowledge $\{q_j^{prior}\}$ we have under our learned policy $\pi_\theta$ as described in Eq. 12.

As shown in Fig. 3, adding verifiers consistently improves performance over using them individually. For $V_1$ = EXP and $V_2$ = MLP (Fig. 3a,b), all three composition strategies (Sec. 4.2) perform similarly, yet each surpasses the single verifiers, underscoring the complementary value of strong verification sources such as experimental data and models of experimental data. When combining knowledge and experimental verifiers, training order is critical. In Fig. 3c,d, models trained with $GO_{all-ll}$ first and then $MLP, EXP$ ($GO_{all-ll}\|MLP, GO_{all-ll}\|EXP$) outperform the reverse. GO-based supervision increases TPR (captures more positives) but lowers TNR; subsequent MLP/EXP training restores TNR, improving Balanced Accuracy and MCC. Starting from MLP or EXP and

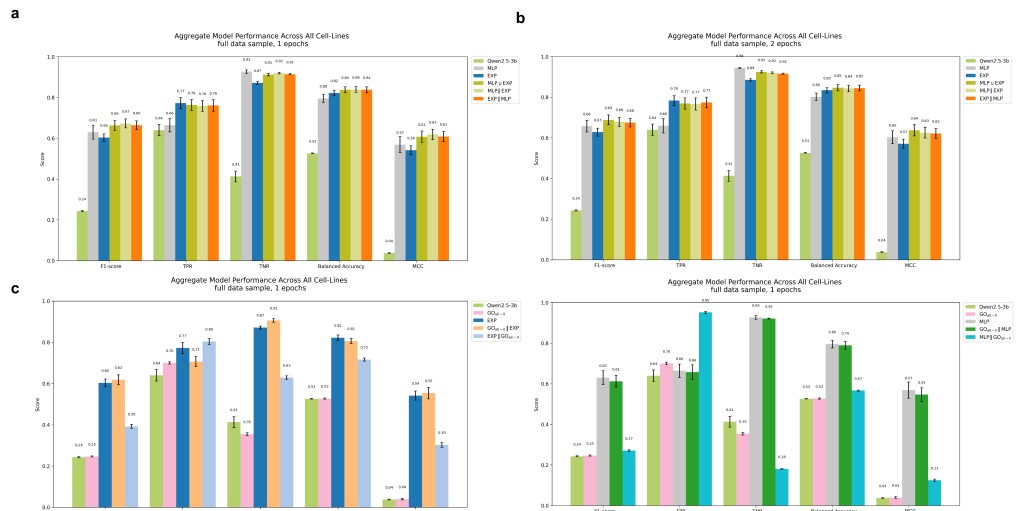

**Figure 3: Model performance for compositions of verifiers** $V_i \| V_j$ corresponds to training models sequentially, first on $V_i$, then on $V_j$. $V_i \cup V_j$ corresponds to models trained on a random mixture of $\{V_i, V_j\}$; **(a, b)** $MLP$ and $EXP$, trained for 1, and 2 epochs. **(c)** $EXP$ and $GO_{all-ll}$ **(d)** $MLP$ and $GO_{all-ll}$

adding $GO_{all-ll}$ later reduces performance, showing that knowledge signals can dilute experimental ones if applied too late. Thus, GO rewards are most useful early for broad guidance, while high-fidelity verifiers refine later. Cross-verifier agreement (**Appendix A.5; SI Figs. 10–12**) shows EXP and MLP are strongly aligned ($r = 0.81$–$0.85$, binary $0.92$–$0.94$). Overall, the results reveal a hierarchy: GO early aids recall, ending with a high-fidelity verifier (GO→MLP/EXP) yields robust signals, while the reverse (MLP/EXP→GO) introduces noise. Later verifiers denoise earlier rewards, indicating effects stem from signal quality rather than reward scaling. This follows a standard training principle—start broad and noisy (ontologies), then refine with higher-quality experimental supervision—and naturally generalizes to multiple verifiers $V_1, V_2, \ldots, V_k$.

### 4.3 RBIO WITH CHAIN-OF-THOUGHT YIELDS STATE OF THE ART ON PERTURBQA

Adding chain-of-thought (CoT) reasoning at inference improves all **rbio1** variants we tested (Table 2), surpassing SUMMER as state-of-art performance on the PerturbQA benchmark. The CoT prompt that performed the best was: *'The Biologist will evaluate each step of this problem, using logical reasoning and evidence from the prompt."* Examples of performance increase: *rbio-EXP-all-cell-lines* F1 $0.75 \rightarrow 0.79$, Balanced Accuracy $0.88 \rightarrow 0.91$, TPR $0.83 \rightarrow 0.87$; *rbio-MLP-ESM* F1 $0.67 \rightarrow 0.71$, Balanced Accuracy $0.85 \rightarrow 0.89$, TPR $0.81 \rightarrow 0.87$. We offer examples of answers and reasoning traces generated by the **rbio1** models on a perturbation question in Figure 5 in Supplementary material. Shown in Figure 4 are **rbio1** models trained on only one-fifth of the data and tested with and without CoT. Remarkably, adding CoT at inference lets them reach state-of-the-art performance on PerturbQA - with *rbio-MLP ∪ EXP-CoT* surpassing SUMMER despite being trained on a fraction of training data - demonstrating the power of inference-time capabilities and verifier composition in reasoning models. Note that here SUMMER's higher TNR reflects a stricter precision–recall trade-off rather than overall superior performance.

### 4.4 RBIO OUTPERFORMS LLMS WITH UP TO 40× MORE PARAMETERS ON PERTURBQA

Despite having only 3 billion parameters, **rbio1** models substantially outperform both reasoning-oriented and instruction-tuned LLMs that are an order of magnitude larger (Table 2). Zero-shot baselines—including DeepSeek R1 distilled models (32B–70B parameters), Qwen2.5 Instruct (3B–72B), and OpenAI o1-series models (20B–120B)—achieve F1 scores between 0.24 and 0.30 and MCC scores below 0.16. In contrast, *rbio*-EXP-CoT reaches F1 = 0.786 and MCC = 0.752, while *rbio*-MLP ∪ EXP-CoT trained on only 1/15 of the data achieves F1 = 0.716 and MCC = 0.668.

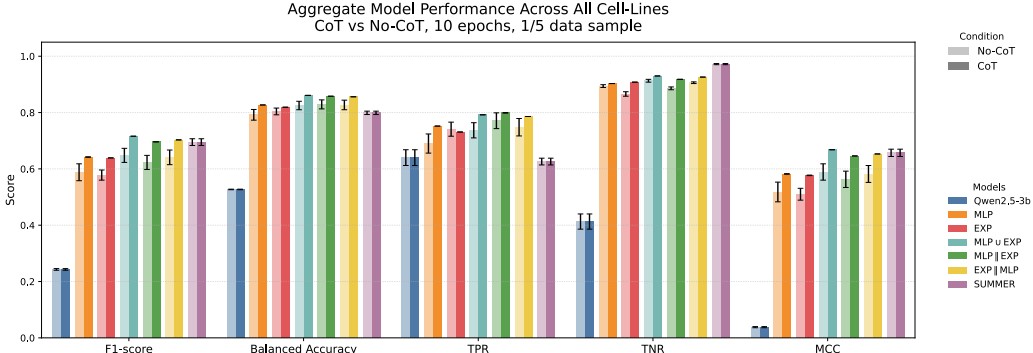

**Figure 4: Effect of chain-of-thought prompting.** Models using CoT achieve state-of-the-art performance on the PerturbQA benchmark.

| Model | F1-score | Balanced Accuracy | TPR | TNR | MCC |
|---|---|---|---|---|---|
| *Models trained on full data size* | | | | | |
| rbio-EXP | $0.750 \pm 0.018$ | $0.883 \pm 0.011$ | $0.827 \pm 0.018$ | $0.939 \pm 0.003$ | $0.709 \pm 0.020$ |
| **rbio-EXP-CoT** | $\mathbf{0.786 \pm 0.000}$ | $\mathbf{0.907 \pm 0.000}$ | $\mathbf{0.872 \pm 0.000}$ | $0.943 \pm 0.000$ | $\mathbf{0.752 \pm 0.000}$ |
| rbio-MLP | $0.669 \pm 0.025$ | $0.855 \pm 0.017$ | $0.807 \pm 0.030$ | $0.902 \pm 0.004$ | $0.618 \pm 0.029$ |
| rbio-MLP-CoT | $0.714 \pm 0.000$ | $0.889 \pm 0.000$ | $0.873 \pm 0.000$ | $0.906 \pm 0.000$ | $0.672 \pm 0.000$ |
| SUMMER | $0.695 \pm 0.012$ | $0.799 \pm 0.006$ | $0.626 \pm 0.012$ | $\mathbf{0.972 \pm 0.002}$ | $0.657 \pm 0.013$ |
| Qwen2.5-3b | $0.231 \pm 0.002$ | $0.522 \pm 0.001$ | $0.529 \pm 0.014$ | $0.515 \pm 0.013$ | $0.032 \pm 0.001$ |
| GEARS | $0.296 \pm 0.033$ | $0.588 \pm 0.012$ | $0.178 \pm 0.024$ | $0.997 \pm 0.001$ | $0.371 \pm 0.023$ |
| *Models trained on 1/15 of full data size* | | | | | |
| rbio-MLP | $0.588 \pm 0.030$ | $0.792 \pm 0.019$ | $0.690 \pm 0.034$ | $0.894 \pm 0.005$ | $0.518 \pm 0.035$ |
| rbio-MLP-CoT | $0.642 \pm 0.001$ | $0.827 \pm 0.000$ | $0.752 \pm 0.000$ | $0.903 \pm 0.000$ | $0.582 \pm 0.001$ |
| rbio-EXP | $0.578 \pm 0.018$ | $0.804 \pm 0.012$ | $0.741 \pm 0.025$ | $0.866 \pm 0.008$ | $0.510 \pm 0.021$ |
| rbio-EXP-CoT | $0.639 \pm 0.000$ | $0.819 \pm 0.000$ | $0.731 \pm 0.000$ | $0.908 \pm 0.000$ | $0.577 \pm 0.001$ |
| rbio-MLP $\cup$ EXP | $0.648 \pm 0.025$ | $0.825 \pm 0.015$ | $0.737 \pm 0.027$ | $0.913 \pm 0.005$ | $0.589 \pm 0.029$ |
| **rbio-MLP $\cup$ EXP-CoT** | $\mathbf{0.716 \pm 0.000}$ | $\mathbf{0.861 \pm 0.000}$ | $0.792 \pm 0.000$ | $0.930 \pm 0.000$ | $\mathbf{0.668 \pm 0.000}$ |
| rbio-MLP $\parallel$ EXP | $0.623 \pm 0.025$ | $0.829 \pm 0.016$ | $0.771 \pm 0.028$ | $0.886 \pm 0.005$ | $0.563 \pm 0.029$ |
| rbio-MLP $\parallel$ EXP-CoT | $0.696 \pm 0.000$ | $0.858 \pm 0.000$ | $\mathbf{0.799 \pm 0.001}$ | $\mathbf{0.918 \pm 0.000}$ | $0.646 \pm 0.000$ |
| rbio-EXP $\parallel$ MLP | $0.641 \pm 0.026$ | $0.827 \pm 0.017$ | $0.748 \pm 0.031$ | $0.906 \pm 0.003$ | $0.582 \pm 0.030$ |
| rbio-EXP $\parallel$ MLP-CoT | $0.703 \pm 0.000$ | $0.856 \pm 0.000$ | $0.786 \pm 0.000$ | $0.926 \pm 0.000$ | $0.653 \pm 0.000$ |
| *Baseline Reasoning (R)/Instruction-tuned models* | | | | | |
| DeepSeek R1 Distil | | | | | |
|   Qwen 32B (R) | $0.241 \pm 0.004$ | $0.512 \pm 0.011$ | $0.694 \pm 0.026$ | $0.340 \pm 0.032$ | $0.024 \pm 0.010$ |
|   Llama 70B (R) † | $0.248 \pm 0.000$ | $0.513 \pm 0.000$ | $0.790 \pm 0.000$ | $0.235 \pm 0.000$ | $0.021 \pm 0.000$ |
| Qwen2.5 3B Instruct | $0.240 \pm 0.003$ | $0.518 \pm 0.012$ | $0.663 \pm 0.026$ | $0.376 \pm 0.030$ | $0.028 \pm 0.010$ |
| Qwen2.5 72B Instruct | $0.247 \pm 0.004$ | $0.543 \pm 0.009$ | $0.517 \pm 0.039$ | $0.569 \pm 0.047$ | $0.045 \pm 0.018$ |
| **OpenAI OSS 20B (R) †** | $\mathbf{0.295 \pm 0.013}$ | $\mathbf{0.602 \pm 0.018}$ | $\mathbf{0.435 \pm 0.036}$ | $\mathbf{0.758 \pm 0.032}$ | $\mathbf{0.151 \pm 0.016}$ |
| OpenAI OSS 120B (R) | $0.289 \pm 0.019$ | $0.602 \pm 0.017$ | $0.279 \pm 0.027$ | $0.896 \pm 0.017$ | $0.131 \pm 0.019$ |

**Table 2: Aggregate PerturbQA performance.** Mean $\pm$ SE over 5 completions across 4 cell lines. *rbio-EXP* corresponds to rbio-EXP-all-cell-lines. Comparison to baselines: SUMMER (Wu et al., 2025) (task SOTA), and SOTA reasoning and instruction-tuned models. Best model in each category bolded. Models with † were unable to answer all prompts. Reasoning/instruction-tuned models evaluated zero-shot.

This demonstrates that domain-specific post-training with soft verification enables smaller models to acquire biological reasoning capabilities that general-purpose LLMs fail to exhibit through pre-training and instruction-tuning alone.

## 5  CONCLUSION

We introduce **rbio1**, a suite of reasoning models trained via *soft verification*, where simulations from biological world models provide rewards for reinforcement learning. This approach rivals experimental-data–trained models, especially when combined with chain-of-thought prompting. By leveraging predictive bio-models (e.g., MLPs on gene embeddings) and knowledge sources like the GO Ontology, **rbio1** shows that simulations and prior knowledge can substitute for costly experimental supervision. We aim to extend **rbio1** across diverse biological models and modalities toward a universal virtual cell system integrating multiple sources as soft verifiers. Soft verification defines a general paradigm for training reasoning LLMs—scalable to domains without exact verifiers and opening new directions for verifier design, noise robustness, and evaluation beyond task accuracy.

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

# A APPENDIX

## A.1 PERTURBATION-REASONING TASK SETUP AND NATURAL-LANGUAGE FORMULATION

**Overview.** The PerturbQA benchmark was introduced by (Wu et al., 2025)and provides CRISPRi single-gene perturbation experiments across four human cell lines—**RPE1**, **K562**, **HEPG2**, and **JU-RKAT**. Each experimental record indicates whether a knock-down of gene $A$ affects the expression of gene $B$ in a specific cell line. We follow the processed natural-language formulation introduced in PerturbQA, using the converted perturbation records into text-based reasoning query.

**Natural-language conversion.** Each example is phrased as a binary scientific question, for instance:

> *"Is a knockdown of AARS in hepg2 cells likely to result in differential expression of ATAD2B? The answer is either yes or no."*

This transformation allows language models to treat biological perturbation prediction as a question-answering task grounded in experimental evidence. Ground-truth labels (yes/no) are derived directly from differential-expression outcomes reported in the original CRISPRi datasets. This representation enables the model to reason jointly over biological entities, relationships, and cell-line context, bridging structured perturbation data with language-based reasoning.

**Prompting formulation.** Each perturbation question is provided as the **user_prompt**, and the model responds according to a fixed **system_prompt** describing a structured reasoning dialogue:

> *"A conversation between User and Biologist. The user asks a question, and the Biologist solves it. The Biologist first thinks about the reasoning process in the mind and then provides the user with the answer. The reasoning process and answer are enclosed within* `<think>` *and* `<answer>` *tags, respectively, i.e.,* `<think>` *reasoning process here* `</think>` `<answer>` *answer here* `</answer>`*."*

This format explicitly separates latent reasoning from the final prediction, allowing reinforcement learning to target biologically grounded reasoning steps and answer correctness.

**Task setups (as illustrated in Fig. 2).** We evaluate three complementary configurations:

- **(a) rbio-EXP-one-cell-line:** Models are trained and evaluated on perturbations within a single cell line (e.g., train = test = RPE1). This setup measures within-cell-line generalization where training and test gene pairs are disjoint, isolating reasoning performance without cross-cell transfer.

- **(b) rbio-EXP-leave-one-out:** Models are trained on three of the four cell lines and tested on the held-out one (e.g., train = RPE1 + K562 + HEPG2 → test = JURKAT). This configuration evaluates out-of-distribution (OOD) generalization across cellular contexts.

- **(c) rbio-MLP-leave-one-out:** In this setting, an MLP surrogate model is first trained on three cell lines using experimental data, then used to generate probabilistic predictions (soft rewards) on the training split of the held-out fourth cell line. Testing is on this split. These model-predicted probabilities serve as reward signals during reinforcement learning, effectively replacing direct experimental supervision.

**Baselines.** We compare against: **SUMMER** (Wu et al., 2025) (retrieval + knowledge-based reasoning), **GEARS** (Roohani et al., 2022) (a specialized perturbation model), and the base **Qwen2.5-3B** reasoning model without biological supervision.

## A.2 METRICS

**Metrics Used** We formulate the genetic perturbation prediction task as a question in natural language with a binary answer. Given a pair of genes $gene_A$ and $gene_B$, the model is asked to emit a binary answer — **yes** or **no**. We use four CRISPRi single-gene perturbation knockdown datasets on four cancer cell lines (RPE1, K562, HEPG2, JURKAT), post-processed into natural language queries by PerturbQA (Wu et al., 2025). We compute the following metrics:

$$Recall\ (TPR) = \frac{TP}{TP + FN} \tag{14}$$

$$TNR = \frac{TN}{TN + FP} \tag{15}$$

$$Precision = \frac{TP}{TP + FP} \tag{16}$$

$$F1\ Score = \frac{2 \cdot Precision \cdot Recall}{Precision + Recall} \tag{17}$$

$$Balanced\ Accuracy = \frac{TPR + TNR}{2} \tag{18}$$

$$MCC\ (Matthews\ Correlation\ Coefficient) = \frac{TP \cdot TN - FP \cdot FN}{\sqrt{(TP + FP)(TP + FN)(TN + FP)(TN + FN)}} \tag{19}$$

**Interpretation.** The PerturbQA datasets are inherently class-imbalanced, with substantially fewer positive (true perturbation) cases than negatives. In this setting, identifying true positive perturbations is biologically more important than avoiding false positives, since missing true regulatory effects (false negatives) can obscure functional gene relationships. We therefore emphasize **Recall (TPR)**, **F1-score**, **Balanced Accuracy**, and **MCC** as the most informative metrics. Recall captures the model's sensitivity to true perturbations, F1 balances precision and recall under imbalance, Balanced Accuracy accounts for both sensitivity and specificity, and MCC provides a single correlation-like summary that remains meaningful even for skewed label distributions. Together, these metrics provide a robust picture of biological prediction fidelity under class imbalance.

**Computation of error bars in reported metrics** For each model $m$ and metric $\ell$, performance values are first aggregated within dataset (corresponding to one of the four cell lines (K562, RPE1, JURKAT, HEPG2) $d \in D$ by taking the mean:

$$\bar{v}_d = \frac{1}{n_d} \sum_{i=1}^{n_d} v_{d,i},$$

where $v_{d,i}$ are per-example scores within dataset $d$. We then compute the mean performance across datasets

$$\bar{v} = \frac{1}{|D|} \sum_{d \in D} \bar{v}_d,$$

and quantify variability across datasets as

$$s = \sqrt{\frac{1}{|D| - 1} \sum_{d \in D} (\bar{v}_d - \bar{v})^2}.$$

The error bars in all aggregate bar plots represent the standard error of the mean (s.e.m.) across datasets:

$$\text{SEM} = \frac{s}{\sqrt{|D|}}.$$

This approach first averages performance within each dataset, then estimates variability across datasets, ensuring that all datasets contribute equally and preventing larger datasets from dominating the uncertainty estimate.

## A.3 EXAMPLES OF REASONING TRACES

*Is a knockdown of ISCA2 in rpe1 cells likely to result in differential expression of CEP295? The answer is either yes or no*

**system_prompt**
A conversation between User and Biologist. The user asks a question, and the Biologist solves it. The biologist first thinks about the reasoning process in the mind and then provides the user with the answer. The reasoning process and answer are enclosed within <think> </think> and <answer> </answer> tags, respectively, i.e., <think> reasoning process here </think> <answer> answer here </answer>.

**system_prompt**
A conversation between User and Biologist. The user asks a question, and the Biologist solves it. The biologist first thinks about the reasoning process in the mind and then provides the user with the answer. The reasoning process and answer are enclosed within <think> </think> and <answer> </answer> tags, respectively, i.e., <think> reasoning process here </think> <answer> answer here </answer>. The Biologist will evaluate each step of this problem, using logical reasoning and evidence from the prompt.

**system_prompt**
You are an AI model trained as a Biologist through reinforcement learning. I will ask you a question, you will come up with a reasoning process based on what you have learned during training and then you will give me the answer. The reasoning process and answer are enclosed within <think> </think> and <answer> </answer> tags, respectively, i.e., <think> reasoning process here </think> <answer> answer here </answer>. You will provide the reasoning step-by-step, using detailed biological knowledge from training.

**Example answer**

*Knockdown of ISCA2 in RPE1 cells would likely lead to differential expression of CEP295. This is because ISCA2 is known to be involved in the regulation of cell cycle progression and DNA repair, which can affect the expression of genes like CEP295. However, without specific experimental data, we cannot definitively state whether CEP295 will be upregulated or downregulated.*

**Example answer using Chain-of-Thought**

*ISCA2 is a protein that is involved in the regulation of cell cycle progression. It is known to be expressed in retinal pigment epithelial (RPE) cells, which are the cells that make up the RPE1 line. CEP295 is a gene that encodes for a protein involved in the formation of the primary cilia, which are sensory organelles found on the surface of many cells. Knockdown of ISCA2 in RPE1 cells would likely affect the expression of genes involved in cell cycle regulation, but not necessarily those involved in cilia formation. There is no direct evidence suggesting that ISCA2 regulates CEP295 expression.*

**Example answer using self-aware Chain-of-Thought**

*Knockdown of ISCA2 in RPE1 cells would likely lead to differential expression of CEP295. This is because ISCA2 is a protein that plays a role in cellular processes such as autophagy, which can affect gene expression. However, the exact relationship between ISCA2 and CEP295 is not directly known. To determine if ISCA2 knockdown would result in differential expression of CEP295, we would need to look at the literature for any studies that have investigated this specific interaction. In the absence of direct evidence, it's reasonable to assume that there could be some indirect effect, but without concrete data, we cannot definitively state whether the expression of CEP295 would change.*

**Figure 5: — Examples of reasoning traces for perturbation prediction, answering a question sampled randomly from the test set.** Each panel shows reasoning text generated within the `<think>` tags (qualitative traces). Different examples correspond to distinct chain-of-thought prompting strategies (standard, explicit chain-of-thought, and self-aware chain-of-thought). Outputs are from a model trained with combinations of soft verifiers. Final answer omitted for brevity

## A.4 ROBUSTNESS TO VERIFIER FIDELITY AND MISCALIBRATION

### A.4.1 EFFECT OF VERIFIER SIGNAL MISCALIBRATION

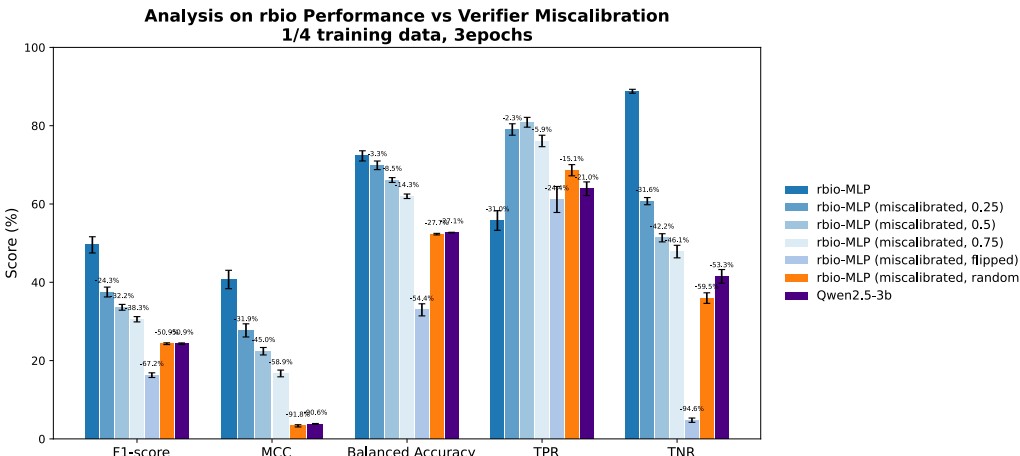

**Figure 6: — Performance of rbio1 under progressively miscalibrated verifier signals.** The fraction of randomized MLP predictions $(0.25 - 0.75)$ or flipped labels is shown on the x-axis, with the extreme case of completely random rewards included. **rbio1** performance (F1, MCC, Balanced Accuracy, TPR, TNR) declines smoothly as verifier noise increases but remains above the base Qwen 2.5-3B model until supervision becomes random, confirming that the model does not amplify verifier errors but instead learns robustly from partial biological signal. Numbers show percentage decrease in performance compared to the model trained on a fully-calibrated MLP.

**Setup.** To assess the robustness of **rbio1** to verifier reliability, we simulate controlled levels of miscalibration in the MLP verifier used during reinforcement learning. During training, each instance receives a scalar reward $r \in [0, 1]$ based on the verifier's predicted probability $p_{\text{yes}}$ (and $p_{\text{no}} = 1 - p_{\text{yes}}$). If the model outputs "yes," the reward is $r = p_{\text{yes}}$; otherwise, $r = p_{\text{no}}$.

To emulate verifier noise, we perturb $p_{\text{yes}}$ according to:

$$p'_{\text{yes}} = \begin{cases} p_{\text{yes}}, & \text{no noise} \\ (1-\rho)\, p_{\text{yes}} + \rho\, U(0,1), & \text{partial randomization, } \rho \in \{0.25, 0.5, 0.75\} \\ 1 - p_{\text{yes}}, & \text{flipped} \\ U(0,1), & \text{fully random} \end{cases} \tag{20}$$

where $U(0,1)$ denotes samples from a uniform distribution over $[0,1)$, implemented with `np.random.rand()`. The complement is set as $p'_{\text{no}} = 1 - p'_{\text{yes}}$. This parameterization allows a smooth transition from correctly calibrated to fully corrupted verifier signals.

**Results.** As shown in Fig. 6, **rbio1** performance (F1, MCC, Balanced Accuracy, TPR, TNR) decreases smoothly as verifier noise increases, but remains well above the Qwen2.5-3B baseline until rewards are completely random. When signals are flipped or randomized, performance converges to—but does not fall below—the base model. This indicates that **rbio1** is able to learn from imperfect verifiers and that training is driven by biologically structured signal rather than incidental correlations.

### A.4.2 REWARD COMPONENTS ABLATIONS

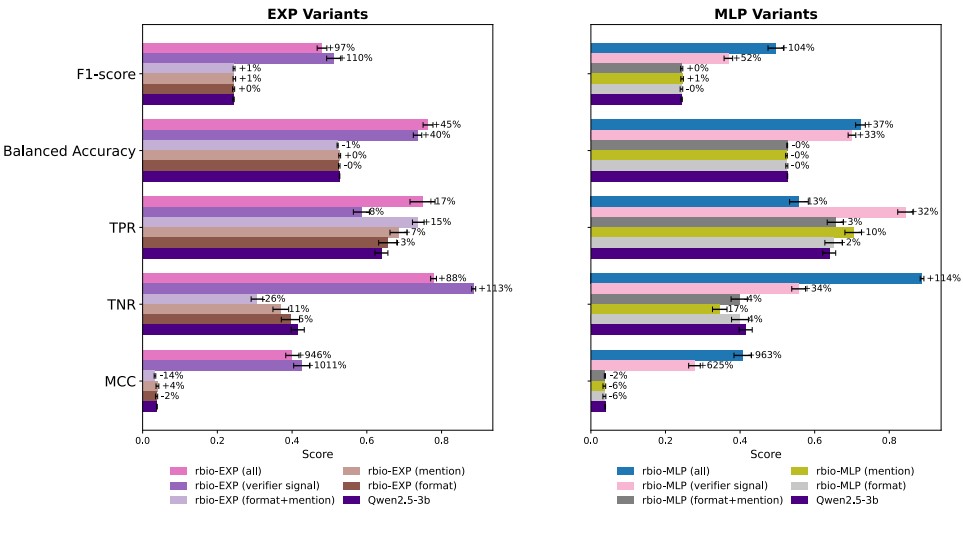

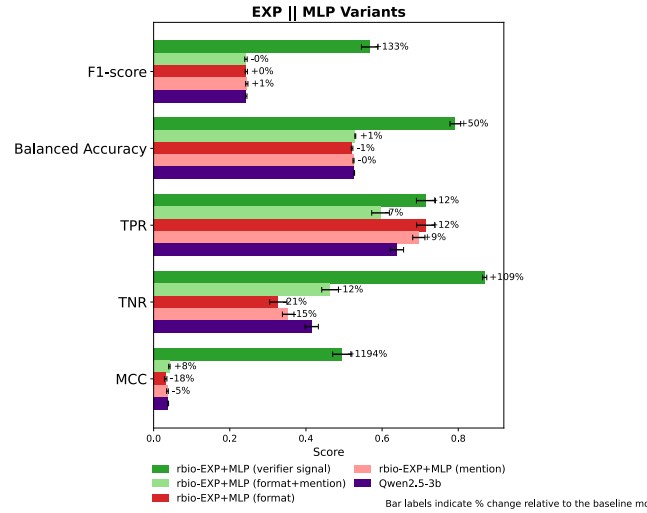

**Figure 7: — Ablation of reward components for rbio1 trained with experimental (EXP), model-based (MLP), and combined (EXP ||MLP) verifiers.** Across all settings, models trained with the full biological-answer reward outperform those using only generic format or mention rewards. The improvement is consistent across metrics and verifier types, indicating that the biological signal—not generic RL regularization—is the dominant contributor to performance.

**Setup.** To isolate the effect of different reward components defined in Eq. 13, we conduct ablations across the three verifier classes—experimental (EXP), model-based (MLP), and combined (EXP ||MLP). The full reward in Eq. 13 can be re-written as:

$$r_i(q, o_i) = r_{\text{format}} + r_{\text{mention}} + r_{\text{verifier}} \tag{21}$$

which includes (i) a *format reward* enforcing structured output, (ii) a *mention reward* encouraging relevant entity inclusion—in our case, gene mentions—and (iii) a *biological-answer reward* $r_{\text{verifier}}$ that encodes signal from verifier $V_k$ (e.g., experimental or MLP-based). We train variants using only individual terms: $r_{\text{format}}$, $r_{\text{mention}}$, $r_{\text{format}} + r_{\text{mention}}$, and the full biological reward $r_{\text{verifier}}$.

**Results.** As shown in Fig. 7, models trained with the full biological-answer reward outperform those using only generic format or mention rewards across all metrics (F1, Balanced Accuracy, TPR, TNR,

MCC) and verifier types. The trend is consistent across EXP, MLP, and EXP ||MLP settings, with gains of up to +100% in F1 and MCC when including $r_{\text{verifier}}$.

These results demonstrate that **rbio1**'s improvements are driven by the *biological signal* encoded in verifier feedback rather than generic RL regularization. Format- or mention-only rewards provide minor stylistic consistency but little biological benefit, whereas the full composition—especially the answer-level reward—contributes substantially to both recall and calibration. This supports our central claim that meaningful scientific rewards, not auxiliary shaping terms, are the dominant source of performance.

**Consistency with generic RL/IT baselines.** Table 2 further supports this conclusion: instruction-tuned and general RL reasoning models (e.g., DeepSeek R1, Qwen Instruct, OpenAI OSS) trained with generic format/helpfulness/self-verification signals achieve only modest performance on PerturbQA, whereas rbio1 variants—differing mainly by the presence of the biological answer reward—substantially exceed these baselines using the same 3B backbone. Together with the ablations above, this indicates that *biological* supervision is the principal driver of improvement, not generic reinforcement or instruction tuning.

### A.4.3 SENSITIVITY TO VERIFIER CONFIDENCE (PER-GENE ANALYSIS)

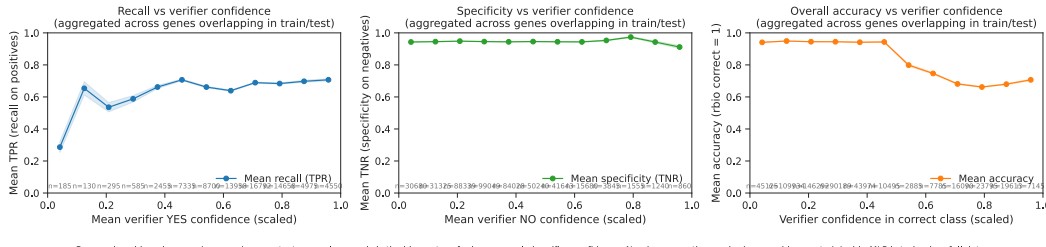

**Figure 8: — Sensitivity of rbio1 performance to verifier fidelity.** Relationship between verifier confidence and **rbio1** test-time performance, aggregated across all perturbed genes present in training and testing. Each point corresponds to the *center of a confidence bin* (after min–max scaling); the y-axis shows the bin mean of the metric with shaded $\pm$ s.e.m. and annotated sample counts $n$. Left: recall (TPR) vs. mean verifier YES-confidence. Center: specificity (TNR) vs. mean verifier NO-confidence. Right: overall correctness (accuracy) vs. verifier confidence in the correct class. **rbio1** recall increases with verifier confidence, while specificity and overall correctness remain high, indicating that the model leverages—but does not depend on—verifier certainty. Performance remains robust even for low-confidence genes, showing resilience to imperfect or miscalibrated verifiers.

**Setup.** Let $g$ index perturbed genes. From training logs, we compute the MLP verifier's mean YES-confidence per gene over all cell lines:

$$\bar{p}_{\text{yes}}(g) = \mathbb{E}_{\text{MLP emissions for } g}[p_{\text{yes}}], \qquad \bar{p}_{\text{no}}(g) = 1 - \bar{p}_{\text{yes}}(g).$$

We left-join these per-gene confidences to each test example $(x, g, y)$ and evaluate the trained policy's binary prediction $\hat{y} \in \{0, 1\}$. For analysis we form:

$$\text{TPR input: } c^+ = \bar{p}_{\text{yes}}(g) \text{ for } y = 1, \quad \text{TNR input: } c^- = \bar{p}_{\text{no}}(g) \text{ for } y = 0, \quad c^\star = \begin{cases} \bar{p}_{\text{yes}}(g) & y = 1 \\ \bar{p}_{\text{no}}(g) & y = 0. \end{cases}$$

Each $c$ is min–max scaled to $[0, 1]$ and partitioned into 12 equal-width bins; the plotted points correspond to bin centers, with y-values equal to binned means of (i) $\mathbb{E}[\hat{y} \mid y=1]$ (TPR), (ii) $\mathbb{E}[1-\hat{y} \mid y=0]$ (TNR), and (iii) $\mathbb{E}[\mathbb{I}\{\hat{y}=y\}]$ (overall correctness).

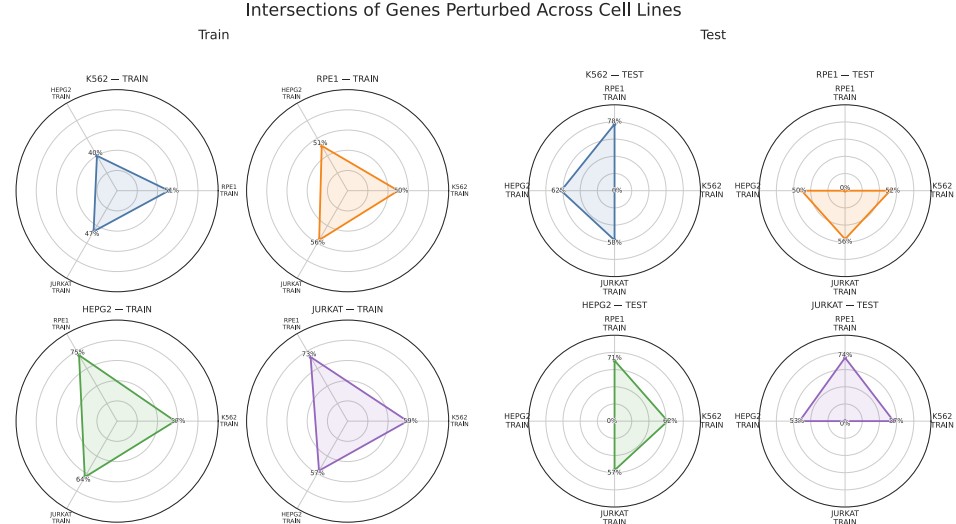

**Figure 9: — Gene overlap across TRAIN sets and TEST.** Radar plots show the percentage of shared perturbed genes between cell lines within training (left) and between training and test splits (right) for each held-out cell line. Each small radar corresponds to one reference cell line and compares its gene set to those of the remaining lines. While cell lines share 40–75% of perturbed genes in training, overlap between a test set and other training sets ranges from 50–78%. This partial but non-trivial intersection indicates that cell lines differ in transcriptional programs yet retain overlapping biological structure, making leave-one-out evaluation both challenging and biologically realistic.

**Connection to cross–cell-line context.** Fig. 9 shows that gene vocabularies are only partially shared across TRAIN splits (40–75%) and between TEST and TRAIN (50–78%), so the sensitivity curves in Fig. 8 reflect aggregation over both shared and distinct gene sets across cell lines rather than a trivially identical vocabulary.

**Results.** As verifier YES-confidence increases, TPR rises monotonically; TNR remains high and relatively flat; and overall correctness (F1 proxy) is stable until the top-confidence tail, where small-sample effects appear. Together with the overlap analysis, this indicates that **rbio1** *leverages* verifier certainty but does not *depend* on it— the model's gains persist under partial gene sharing across cell lines and remain robust to verifier uncertainty.

## A.5    VERIFIER AGREEMENT AND COMPOSITION EFFECTS

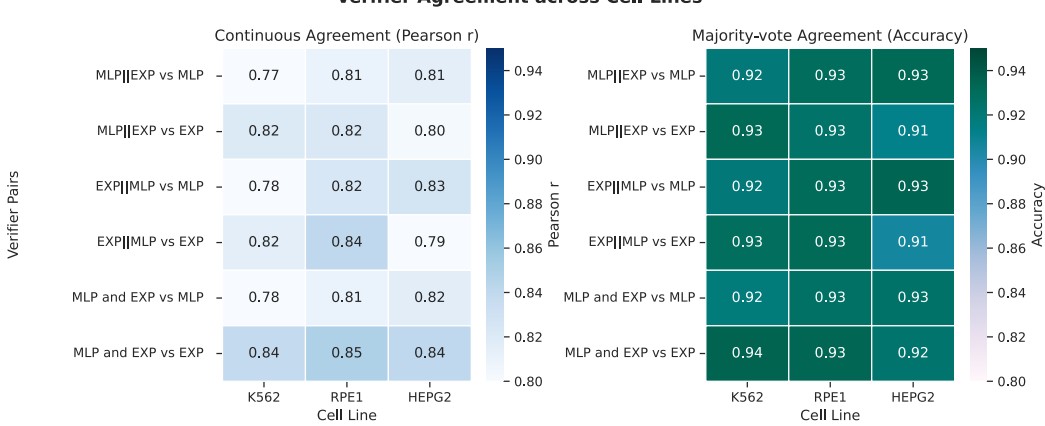

**Figure 10: — Verifier agreement across cell lines for MLP and EXP verifiers.** Agreement measured as Pearson correlation of continuous scores (left) and majority-vote agreement (right) between verifier outputs.

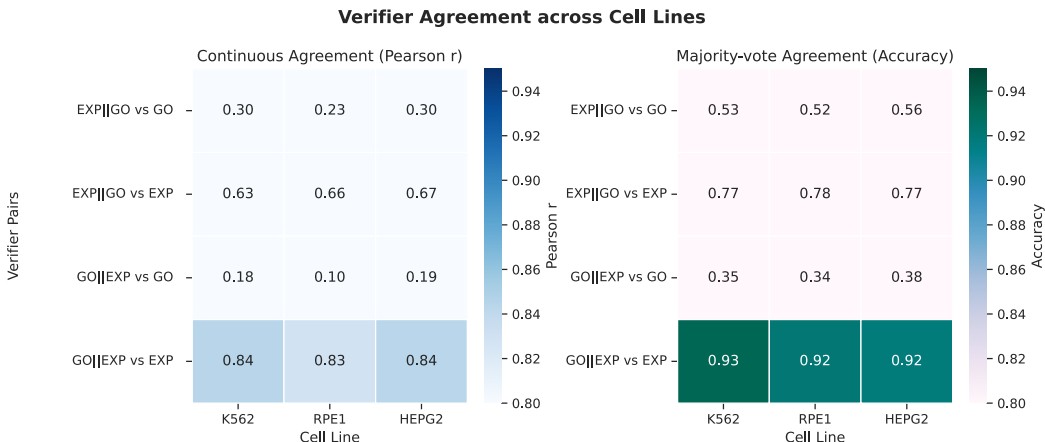

**Figure 11: — Verifier agreement across cell lines for GO and EXP verifiers.** Agreement measured as Pearson correlation of continuous scores (left) and majority-vote agreement (right) between verifier outputs.

**Figure 12: — Verifier agreement across cell lines for GO and MLP verifiers.** Agreement measured as Pearson correlation of continuous scores (left) and majority-vote agreement (right) between verifier outputs.

**Setup.** We compare verifier outputs across the four PerturbQA cell lines (K562, RPE1, HEPG2, and Jurkat) to quantify consistency between the experimental (EXP), model-based (MLP), and ontology-derived (GO) verifiers. For each cell line, we compute (i) the *continuous agreement*, measured as the Pearson correlation $r$ between the continuous verifier scores; and (ii) the *majority-vote agreement*, measured as binary accuracy between thresholded verifier predictions. These statistics are aggregated over all shared gene pairs within each cell line.

**Results.** As shown in Figs. 10–12, EXP and MLP verifiers exhibit high agreement across all cell lines ($r \approx 0.8$, binary agreement $\approx 0.93$), indicating that both encode consistent biological signal. By contrast, GO-based verifiers show weaker raw correlation with EXP or MLP ($r \approx 0.3$) but still moderate binary alignment (accuracy $\approx 0.75$), reflecting that ontology-derived priors capture complementary but coarser relationships. Compositional verifiers such as GO ||EXP and GO ||MLP realign more closely with the higher-fidelity verifiers applied last, confirming that the order of composition influences which signal dominates.

**Interpretation.** High pairwise consistency among EXP and MLP verifiers supports that **rbio1** learns from largely aligned supervision sources rather than conflicting signals. The weaker continuous correlation but stable discrete alignment of GO-based rewards suggests that these verifiers contribute structured regularization rather than direct label imitation—providing a complementary prior that the reinforcement process integrates effectively across cell lines.

## A.6 REINFORCEMENT LEARNING VS. SUPERVISED VERIFIERS

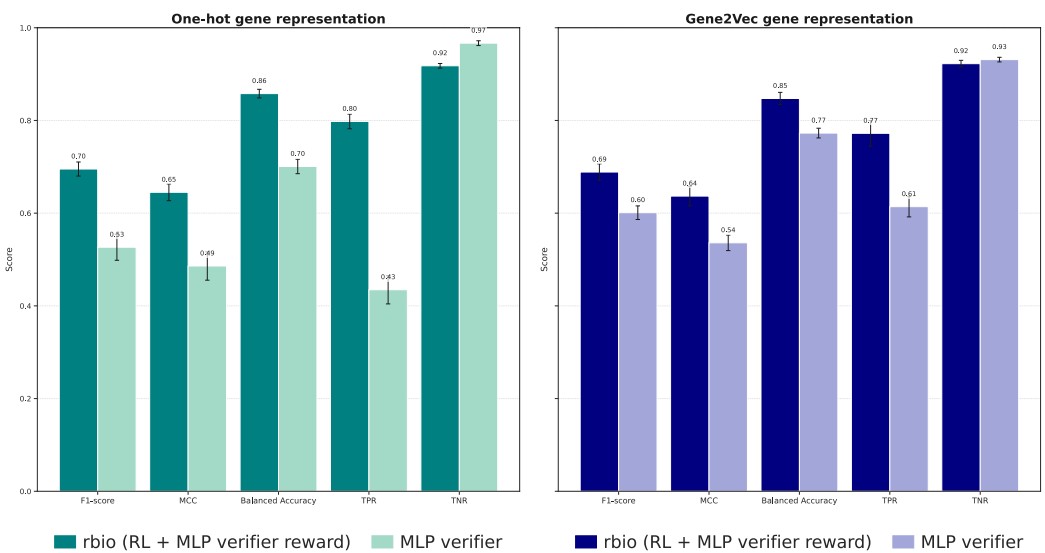

Figure 13: — **Performance of the MLP verifier compared with rbio1 models trained with the MLP as a soft verifier. rbio1** models were trained on 1/4 of the data for 3 epochs, using chain-of-thought (CoT) reasoning at inference. Left: MLP trained with one-hot gene representations. Right: MLP trained with Gene2Vec representations. The MLP decision threshold was set to 0.5 for positive interactions. Across all metrics, **rbio1=(RL + MLP verifier reward + CoT)** outperforms the MLP verifier, indicating that reinforcement learning contributes beyond supervised imitation and does not amplify verifier noise. The two rbio1 models showcased correspond to *rbio-MLP-leave-one-out-one-hot* and *rbio-MLP-leave-one-out-gene2vec*

### A.6.1 MLP ARCHITECTURE AND TRAINING

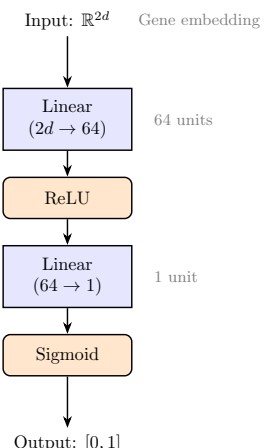

Figure 14: MLP Structure

Table 3: MLP Training Hyperparameters

| Hyperparameter Name | Value |
|---|---|
| Number of Epochs | 10 |
| Batch Size | 32 |
| Learning Rate | $1 \times 10^{-3}$ |
| Random Seed | 42 |

**Setup.** The MLP verifier is a simple two-layer network (Fig. 14) consisting of a linear projection from the concatenated gene embedding ($\mathbb{R}^{2d} \to 64$), followed by a ReLU activation, a second linear layer ($64 \to 1$), and a sigmoid output. It produces a scalar confidence $p_{\text{yes}} \in [0, 1]$ representing the probability of a positive perturbation effect. Training hyperparameters are listed in Table 3. We evaluate two gene-encoding schemes:

- **One-hot representation:** genes are represented by binary one-hot vectors, providing no relational prior.

- **Gene2Vec embedding:** genes are embedded in a dense continuous space learned from large-scale co-expression networks, capturing semantic relationships between genes.

**Results and significance.** As shown in Fig. 13, **rbio1** models trained using the MLP as a soft verifier (via reinforcement learning under the GRPO objective) outperform the MLP verifier itself across all metrics (F1, MCC, Balanced Accuracy, TPR, TNR) and for both input representations. The largest gains occur in recall (TPR), indicating that the reinforcement learning step allows the model to generalize beyond the fixed decision boundary of the MLP. While the MLP provides the biological reward signal, the reinforcement objective enables the policy to explore multiple reasoning trajectories at inference (via CoT) and to refine predictions through multi-sample consistency. This demonstrates that **rbio1** benefits not only from biological supervision but also from reinforcement optimization that integrates reasoning dynamics and soft-verifier feedback—allowing improvements that pure supervised imitation cannot achieve.

## A.7 OUT-OF-DISTRIBUTION AND CROSS-DOMAIN GENERALIZATION

**Figure 15: — Zero-Shot Disease prediction Performance on Alzheimer and Cancer datasets using rbio1 models trained with models of perturbation data.** Models are trained on a fraction of the available data (top-100 highly expressed genes + metadata). We compare baseline Qwen2.5-3B, SCVI, and **rbio1** variants using various verifiers: rbio-EXP, rbio-MLP, rbio-MLP ∪ EXP and rbio-MLP ∪ EXP ∪ GO. Chain-of-thought was solicited at inference time. Across both datasets, **rbio1** substantially improves recall and F1 over the Qwen2.5-3b baseline and approaches SCVI despite using only a data fraction, highlighting that soft-verifier RL generalizes beyond perturbation prediction to a distinct disease-classification task.

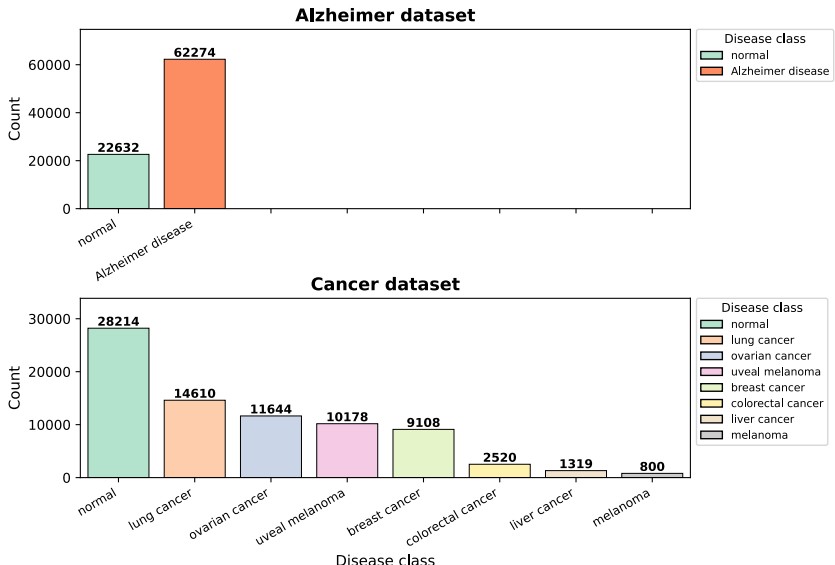

**Figure 16: — Dataset Class Distribution for Disease Prediction Task.** The Alzheimer dataset has two binary labels and the Cancer dataset has seven different types of cancer as cell states. Both datasets have been obtained from CELLXGENE

**Dataset composition.** We evaluate generalization beyond perturbation reasoning on two disease-state datasets: (i) **Alzheimer's disease**, a binary classification task (normal vs. diseased; Fig. 16, top), and (ii) **Myeloid cancers**, a multi-class task containing eight disease states (normal plus seven cancer types; Fig. 16, bottom). Each cell or sample is represented using the top 100 highly expressed genes and relevant experimental metadata, including tissue, assay type, developmental stage, cell type, and organism. All gene-expression data are sourced from CELLxGENE public collections.

**Prompt construction and input representation.** For each single-cell observation, we generate a natural-language query that integrates biological metadata and expression context. Each prompt follows the format:

> *"This is a single-cell observation obtained using [assay] from the [tissue] of a [sex] Homo sapiens. The top 100 expressed genes in this cell are: [$gene_1$, $gene_2$, ..., $gene_{100}$]. Based on these genes, identify the disease state of this single cell. Choose one: [$class_1$ | $class_2$ | ... | $class_n$]."*

For example:

> *"This is a single-cell observation obtained using 10x 3' v2 from the ovary of an unknown female Homo sapiens. The top 100 expressed genes in this cell are: MALAT1, TMSB4X, B2M, ACTB, RPL10, MT-ATP6, ..., LY6H. Based on these genes, identify the disease state of this single cell. Choose one: normal | lung cancer | ovarian cancer | uveal melanoma | colorectal cancer | breast cancer | liver cancer | melanoma."*

This structured input provides biologically interpretable features that the language model can reason over. Metadata tokens give high-level context, while ranked genes reflect the underlying transcriptional state. Together, they form a natural-language interface between single-cell biology and language-based reasoning.

**Zero-shot evaluation.** All models are evaluated in a zero-shot setting—no fine-tuning or disease-specific supervision is used. **rbio1** models are trained only on perturbation-prediction tasks from PerturbQA using biological soft-verifier rewards (MLP, EXP, and GO). The disease tasks thus test transfer to an out-of-distribution setting, assessing whether models trained on causal perturbations can generalize to disease-state reasoning.

**Example model response.** Below is an example of an **rbio1** reasoning trace, illustrating how the model integrates gene-expression patterns and biological context to infer cell identity and disease state:

> `<think>` The expression levels of different genes can give us an idea of the cell type. For example, genes related to ribosomal proteins are often expressed at high levels in cells that are actively dividing or have a high rate of protein synthesis. Given that there are multiple genes for ribosomal proteins (RPL10, RPL41, RPL13A, RPL13, RPL7, RPL28, RPL37A, etc.), as well as other genes involved in translation (such as EEF1A1, EEF1A2), we can infer that this cell is likely a rapidly proliferating cell. MALAT1 is known to be associated with the epithelial-mesenchymal transition and metastasis, which may suggest the presence of a tumor. B2M (beta-2 microglobulin) is a protein marker for antigen presentation and can be found on the surface of many types of immune cells and some tumor cells. Vimentin (VIM) is a marker of mesenchymal cells and can also be found in some tumors. All of these factors could indicate a potential tumor cell state. However, since the gene list does not explicitly mention any of the known markers for specific cancers, we must consider the most probable option given the data available. The presence of ribosomal proteins and genes related to translation and the possibility of a tumor cell state points towards ovarian cancer as the most plausible diagnosis. `</think>`
> `<answer>` ovarian cancer `</answer>`

**SCVI baseline.** As a reference, we train a standard **SCVI** model directly on the same Alzheimer and Cancer datasets. Following the default training setup:

1. SCVI is initialized using `scvi.model.SCVI` with the disease label as `labels_key="disease"`.

2. The model is trained for 100 epochs with learning rate $1 \times 10^{-3}$.

3. The latent representation is extracted and used to train a linear logistic regression classifier for disease-state prediction.

The model's mean performance is computed from accuracy, precision, recall, and F1 score as in the main evaluation pipeline.

**Table 4:** SCVI training and evaluation setup.

| Parameter | Value / Description |
|---|---|
| Model | `scvi.model.SCVI` (default) |
| Training epochs | 100 |
| Learning rate | $1 \times 10^{-3}$ |
| Latent dimension | default (10) |
| Classifier | Logistic Regression ($L_2$ regularization, max_iter=200) |
| Evaluation metrics | Accuracy, Precision, Recall, F1 (macro) |
| Input data | Same Alzheimer and Cancer datasets as **rbio1** experiments |

**Results and interpretation.** As shown in Fig. 15, **rbio1** achieves large gains over EXP and MLP verifiers across metrics (F1, precision, recall) while maintaining calibration. On the Alzheimer dataset, **rbio1** doubles the F1 score and achieves a $+136\%$ increase in recall relative to baseline verifiers. For Myeloid cancers, **rbio1** improves F1 by 30–70% while retaining high specificity. These results approach those of SCVI—a specialized expression model trained directly on raw counts—despite **rbio1** being trained exclusively on perturbation reasoning signals.

**Significance.** This experiment demonstrates that **rbio1** internalizes generalizable molecular reasoning patterns. Trained solely with reinforcement learning from biological verifiers, it transfers this understanding to infer disease states in unseen data distributions. This zero-shot ability highlights the potential of soft-verifier RL to unify experimental, model-based, and knowledge-based biological reasoning into a single transferable system.

## A.8 DATA AND COMPUTE SCALING

**Setup.** We systematically vary two axes of training—dataset size and compute—to examine scaling behavior and robustness of **rbio1** under different supervision regimes. Each variant uses the same Qwen2.5-3B-Instruct base model trained under the GRPO objective with identical hyperparameters. Dataset-size experiments sample 20%, 50%, and 100% of the full PerturbQA training data across verifier types (EXP, MLP, EXP ∪ MLP, and EXP ∪ MLP ∪ GO) and are run for 1, 3, or 5 epochs. Compute experiments fix the full dataset and vary the number of optimization steps (1, 3, and 5 epochs equivalent) to test performance scaling with training duration.

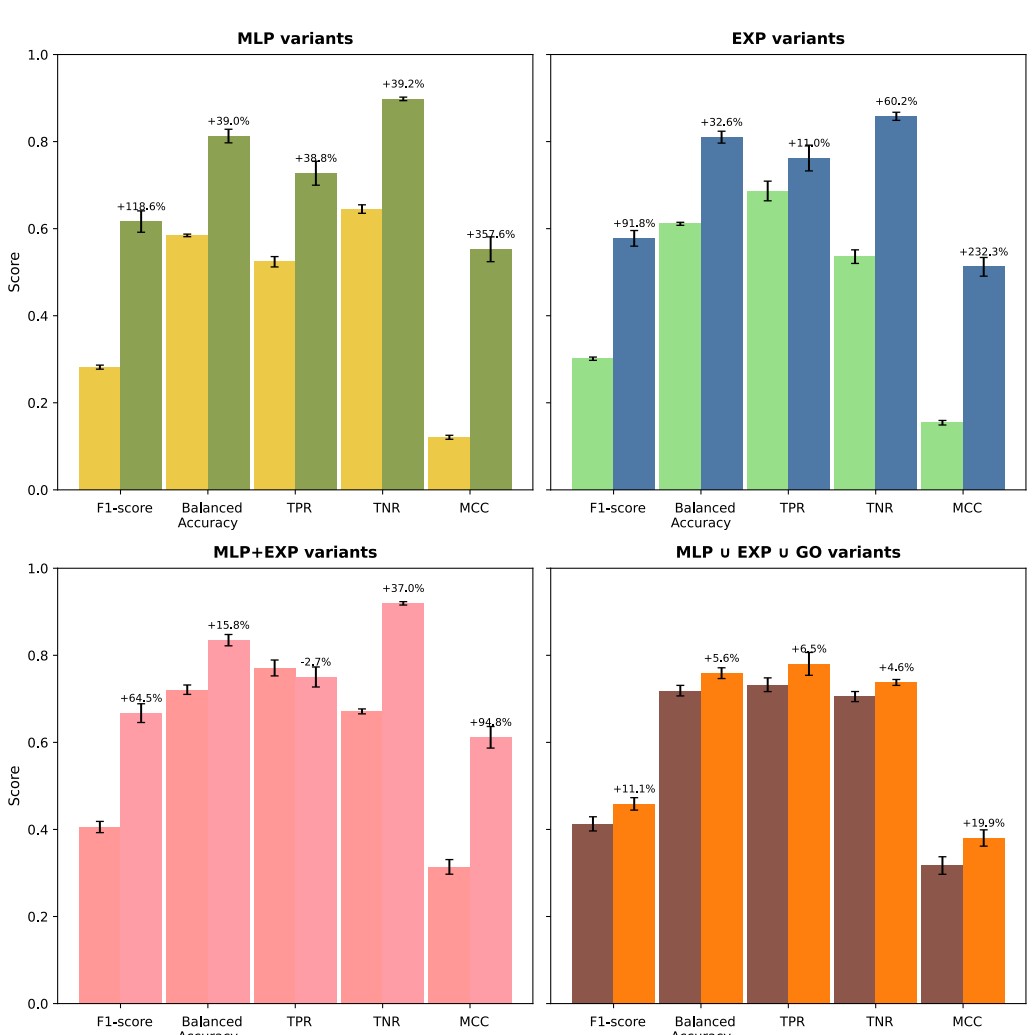

Figure 17: — Effect of training data size on rbio1 accuracy and generalization, 1 epoch.

**Results — data scaling.** Across all verifier configurations and metrics (F1, Balanced Accuracy, TPR, TNR, MCC), performance increases predictably with the amount of training data (Figs. 17–19). The scaling trend is approximately log-linear, with diminishing returns at higher data fractions but consistent improvement in both sensitivity and specificity. This indicates that the GRPO-based optimization effectively captures additional signal as more examples are available.

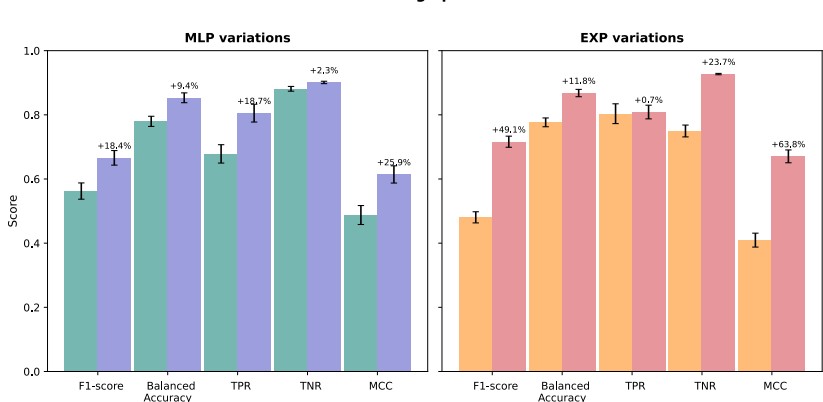

**Figure 18: — Effect of training data size on rbio1 accuracy and generalization, 3 epochs.**

**Figure 19: — Effect of training data size on rbio1 accuracy and generalization, 5 epochs.**

## A.9 COMPUTE EFFICIENCY

**Results — compute scaling.** Increasing training compute yields analogous improvements (Fig. 20). Metrics such as F1 and MCC rise smoothly with training steps, confirming that **rbio1** adheres to reinforcement-learning scaling laws observed in other large-model settings.

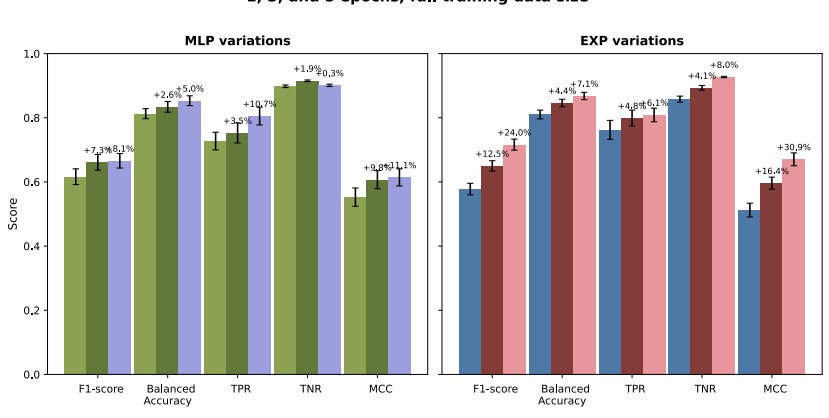

**Figure 20: — Influence of training compute at 1, 3, and 5 epochs on rbio1 performance.**

**Interpretation.** Together, these analyses show that **rbio1** scales predictably with both data and compute—doubling training data or epochs consistently improves recall, precision, and calibration. This behavior demonstrates that the biological reward signals provide a smooth and information-rich learning gradient, enabling steady performance growth without reward collapse. It further confirms that reinforcement learning over biological verifiers is a robust and scalable paradigm for large-model training in scientific domains.

## A.10 TRAINING AND EVALUATION

Models have been trained using the GRPO framework and the HuggingFace interface. We use a Qwen2.5-3B-Instruct model as a base model, accessed through HuggingFace. Most models presented on individual verifiers have been trained for up to 100k steps with `max_completion_len=256`, taking approximately 10 days to complete on 8 H100 GPUs. Models containing compositions of verifiers were trained for up to 10 epochs. Ablation and sensitivity experiments used proportionally shorter runs or reduced data fractions to ensure efficiency while preserving relative comparisons. Each experiment presents the checkpoint corresponding to its stated data fraction and number of training epochs; while absolute values may vary, the observed performance trends are consistent across configurations. All models used `batch_size=4`, `n_generation=4`, and a learning rate of $5 \times 10^{-6}$. During inference, each model was prompted for `N=5` generations with `max_new_tokens=1024`, `do_sample=True`, `temperature=0.7`, `top_p=0.9`, `top_k=50`. Metrics are reported over five different generations. Each model also includes formatting rewards, following Guo et al. (2025).

**Table 5:** Training configuration for **rbio1** models.

| Model Specification | |
|---|---|
| Model | Qwen2.5-3B-Instruct (3.09B parameters) |
| **Training Hyperparameters** | |
| Per-device train batch size | 4 |
| Number of rollouts | 4 |
| Max completion length | 256 |
| Length penalty | 200 |
| Learning rate | $1 \times 10^{-6}$ |
| Random seed | 42 |

### A.10.1 DATA AND CODE AVAILABILITY

We used the pre-processed versions, as well as the training and testing splits of the perturbation datasets on the four cell lines (K562, RPE1, HEPG2, JURKAT) from https://github.com/genentech/PerturbQA.

We have released an anonymous repository (`https://anonymous.4open.science/r/rbio-9155/README.md`) that reproduces the MLP-verifier experiments and provides an end-to-end example of training an *rbio* model using the MLP signal. This release focuses on the essential components for reproducibility and community adoption. A public, non-anonymized release will follow post-publication and will include all code, checkpoints, and datasets used in the paper. We are fully committed to open science and long-term reproducibility.

Model weights will also be made publicly available upon publication. For the disease prediction task, the datasets were obtained from CELLXGENE with the following identifiers: Alzheimer Dataset: https://cellxgene.cziscience.com/collections/0d35c0fd-ef0b-4b70-bce6-645a4660e5fa and Cancer Dataset: https://cellxgene.cziscience.com/collections/3f7c572c-cd73-4b51-a313-207c7f20f188.

## A.11 ALGORITHMS

---

**Algorithm 1** Rbio-RLEXP: Reinforcement Learning with Hard Verification

---

**Require:** Dataset of prompts/experimental outcome labels $\{X_i, Y_i\}_{i=1}^N$
**Require:** Model parameters $\theta$ implementing policy $\pi_\theta$
**Require:** Hyperparameters: $\tau$ (temperature), $G$ (generations per prompt), $\beta$ (KL penalty), $\epsilon$ (clipping)
**Ensure:** Trained model with parameters $\tilde{\theta}$ implementing policy $\pi_{\tilde{\theta}}$
1: Initialize $\theta$ from supervised fine-tuned LLM
2: **for** each step $t = 1$ to $T$ **do**
3:     Sample batch indices $b \subset \{1, \ldots, N\}$ uniformly at random
4:     Retrieve batch $\{X_b, Y_b\}$ from dataset
5:     **for** each prompt $X_b$ in batch **do**
6:         **for** $i = 1$ to $G$ **do**
7:             Generate sequence $o_i \sim \pi_\theta(\cdot \mid X_b)$ using model $\theta$ and policy $\pi_\theta$
8:             Extract binary answer $\hat{a}_i$ from $o_i$ (if existing)
9:             **if** answer $\hat{a}_i$ exists **then**
10:                 Score against ground truth $Y_b$:
11:                 **if** $\hat{a}_i = Y_b$ **then**
12:                     $r_i^{\text{hard}} = 1$
13:                 **else**
14:                     $r_i^{\text{hard}} = 0$
15:                 **end if**
16:             **else**
17:                 $r_i^{\text{hard}} = 0$ {Penalize missing answer}
18:             **end if**
19:             Add auxiliary rewards: $r_i = r_i^{\text{hard}} + r_{\text{format}} + r_{\text{mention}}$
20:         **end for**
21:     **end for**
22:     Compute normalized advantages $\hat{A}_{i,t}$ using Eq. (4)
23:     Update $\theta$ via GRPO objective (Eq. 2) with KL divergence penalty (Eq. 5)
24: **end for**
25: **return** $\tilde{\theta}$

---

---

**Algorithm 2** Rbio-RLEMF: Reinforcement Learning with Experimental Model Feedback

---

**Require:** Dataset of prompts $\{X_i\}_{i=1}^N$ (without experimental labels)
**Require:** Pre-trained frozen model $\Phi$ (e.g., MLP, VCM)
**Require:** Model parameters $\theta$ implementing policy $\pi_\theta$
**Require:** Reward transformation function $\eta$: maps model predictions to rewards in $[0,1]$
**Require:** Hyperparameters: $\tau$ (temperature), $G$ (generations per prompt), $\beta$ (KL penalty), $\epsilon$ (clipping)
**Ensure:** Trained model with parameters $\tilde{\theta}$ implementing policy $\pi_{\tilde{\theta}}$
  1: Initialize $\theta$ from supervised fine-tuned LLM
  2: **for** each step $t = 1$ to $T$ **do**
  3:     Sample batch indices $b \subset \{1, \ldots, N\}$ uniformly at random
  4:     Retrieve batch $\{X_b\}$ from dataset
  5:     **for** each prompt $X_b$ in batch **do**
  6:       **for** $i = 1$ to $G$ **do**
  7:         Generate sequence $o_i \sim \pi_\theta(\cdot \mid X_b)$ using model $\theta$ and policy $\pi_\theta$
  8:         Extract binary answer $\hat{a}_i$ from $o_i$ (if existing)
  9:         **if** answer $\hat{a}_i$ exists **then**
10:           Query frozen model: $\hat{p} = \Phi(X_b)$ {Model prediction}
11:           Transform prediction to reward: $r_i^{\text{soft}} = \eta(\hat{p}, \hat{a}_i) \in [0,1]$
12:         **else**
13:           $r_i^{\text{soft}} = 0$ {Penalize missing answer}
14:         **end if**
15:         Add auxiliary rewards: $r_i = r_i^{\text{soft}} + r_{\text{format}} + r_{\text{mention}}$
16:       **end for**
17:     **end for**
18:     Compute normalized advantages $\hat{A}_{i,t}$ using Eq. (4)
19:     Update $\theta$ via GRPO objective (Eq. 2) with KL divergence penalty (Eq. 5)
20: **end for**
21: **return** $\tilde{\theta}$

---

---

**Algorithm 3** Rbio-RLPK: Reinforcement Learning from Prior Knowledge

---

**Require:** Dataset of prompts $\{X_i\}_{i=1}^N$ (without labels)
**Require:** Knowledge source $KS$ (e.g., Gene Ontology)
**Require:** Model parameters $\theta$ implementing policy $\pi_\theta$
**Require:** Knowledge scoring function $\nu$: scores reasoning traces against prior knowledge
**Require:** Hyperparameters: $\tau$ (temperature), $G$ (generations per prompt), $\beta$ (KL penalty), $\epsilon$ (clipping)
**Ensure:** Trained model with parameters $\tilde\theta$ implementing policy $\pi_{\tilde\theta}$
  1: Initialize $\theta$ from supervised fine-tuned LLM
  2: **for** each step $t = 1$ to $T$ **do**
  3:    Sample batch indices $b \subset \{1, \dots, N\}$ uniformly at random
  4:    Retrieve batch $\{X_b\}$ from dataset
  5:    **for** each prompt $X_b$ in batch **do**
  6:        Query knowledge source: $\{q_j^{\text{prior}}\} = \texttt{query\_KS}(X_b)$ {Retrieve relevant prior knowledge}

  7:        **for** $i = 1$ to $G$ **do**
  8:           Generate sequence $o_i \sim \pi_\theta(\cdot \mid X_b)$ using model $\theta$ and policy $\pi_\theta$
  9:           Extract gene_information $o_i^{\text{relevant}}$ from $o_i$ (from `<gene>` tags)
10:           **if** gene_information $o_i^{\text{trace}}$ exists **then**
11:               Score gene_information against prior knowledge: $r_i^{\text{soft}} = \nu(o_i^{\text{trace}}, \{q_j^{\text{prior}}\})$
12:           **else**
13:               $r_i^{\text{soft}} = 0$ {Penalize missing gene information}
14:           **end if**
15:           Add auxiliary rewards: $r_i = r_i^{\text{soft}} + r_{\text{format}} + r_{\text{mention}}$
16:        **end for**
17:    **end for**
18:    Compute normalized advantages $\hat{A}_{i,t}$ using Eq. (4)
19:    Update $\theta$ via GRPO objective (Eq. 2) with KL divergence penalty (Eq. 5)
20: **end for**
21: **return** $\tilde\theta$

---

**Algorithm 4** Formatting Reward $r_{\text{format}}$

---

**Require:** Completion $o_i$
**Require:** Set of formatting constraints $\mathcal{F} = \{F_1, F_2, \dots, F_k\}$
**Ensure:** Formatting reward $r_{\text{format}} \in [0, 1]$
  1: Initialize score vector $s = []$
  2: **for** each constraint $F_j \in \mathcal{F}$ **do**
  3:    **if** $F_j$ is satisfied in $o_i$ **then**
  4:        Append $1.0$ to $s$
  5:    **else**
  6:        Append $0.0$ to $s$
  7:    **end if**
  8: **end for**
  9: $r_{\text{format}} = \frac{1}{|\mathcal{F}|} \sum_{j=1}^{|\mathcal{F}|} s_j$
10: **return** $r_{\text{format}}$

---

---

**Algorithm 5** Mention Reward $r_{\text{mention}}$

---

**Require:** Completion $o_i$
**Require:** Set of desired terms $\mathcal{T} = \{t_1, t_2, \ldots, t_m\}$
**Ensure:** Mention reward $r_{\text{mention}} \in [0, 1]$
  1: Extract reasoning trace $o_i^{\text{trace}}$ from $o_i$ (from `<think>` tags)
  2: Initialize score vector $s = []$
  3: **for** each term $t_j \in \mathcal{T}$ **do**
  4:     **if** $t_j$ appears in $o_i^{\text{trace}}$ **then**
  5:         Append $1.0$ to $s$
  6:     **else**
  7:         Append $0.0$ to $s$
  8:     **end if**
  9: **end for**
 10: $r_{\text{mention}} = \frac{1}{|\mathcal{T}|} \sum_{j=1}^{|\mathcal{T}|} s_j$
 11: **return** $r_{\text{mention}}$

---

