# OpenReview forum: "rbio1 - training scientific reasoning LLMs with biological world models as soft verifiers"
_ICLR.cc/2026/Conference — ICLR 2026 Conference Desk Rejected Submission_

### Official Review · Reviewer_AHen · 2025-10-24

**Soundness:** 3
**Presentation:** 2
**Contribution:** 4
**Rating:** 6
**Confidence:** 2

**Summary:**

This paper introduces rbio1, a reasoning model for biology trained using reinforcement learning with soft verification as a training signal for scientific reasoning LLMs in domains (such as biology) where ground-truth, executable verifiers are scarce. The authors propose RLEMF (reinforcement learning with experimental model feedback) and RLPK (reinforcement learning from prior knowledge) using curated databases such as Gene Ontology. Empirically, the approach is evaluated on perturbation prediction tasks from the PerturbQA benchmark, demonstrating that rbio1 surpasses the current state-of-the-art metrics.

**Strengths:**

The paper presents a novel and highly interesting approach to training scientific reasoning models. It proposes methods for training reasoning in scientific domains that lack formal verifiers. The method is relatively convincing and promising, both practically and empirically. The distinction between RLEMF and RLPK provides a well-structured framework for different types of soft verification. Empirically, the model outperforms current baselines, and the leave-one-out evaluation across cell lines provides good evidence of out-of-distribution generalisation.

**Weaknesses:**

1. There are grammatical errors and typos in the paper. I would suggest rereading it carefully. For example:
– “presemt” → “present” on line 19
– “QWEN Team” → “Qwen Team” on line 45

2. While the methodology seems promising, the computational resources required are not reported in the paper. Is the training generally computationally expensive?

3. The paper does not thoroughly analyse what the soft verifiers are learning, or how their prediction quality affects downstream reasoning performance. It would be helpful to elaborate further on this.

4. While the baselines are comprehensive, the paper could include stronger reasoning baselines (e.g., other RL-trained reasoning models using the same backbone).

**Questions:**

1. GEARS performs surprisingly poorly (F1 = 0.296), which seems inconsistent with published results. Is there any explanation or analysis regarding this?

2. Could you elaborate more on the hyperparameters, computational requirements, seeds, and other details of the training process?

3. How well does the approach generalise? For instance, have you tested it on other perturbation datasets beyond PerturbQA?

4. Could you provide a more complete learning curve showing how performance scales from 1/15 to the full dataset?

5. What are the limitations and failure cases? Could you provide some examples?

---

> ### Author Response · Authors · 2025-11-23
> **Response to Reviewer AHen**
>
> We thank the reviewer for their positive and constructive feedback and for recognizing the novelty and promise of rbio in enabling scientific reasoning from soft verifiers.
>
> **Compute and training details.** We added a Training & Evaluation section in SI A.10 specifying all hyperparameters, compute requirements, and experimental scales. Most models were trained for up to 100 k steps, with smaller-scale experiments explicitly reported when using fewer steps or less data. Training used max_completion_len=256 on 8×H100 GPUs for ~10 days, with batch_size=4, learning_rate=5e-6, and n_generation=4. Evaluation used max_new_tokens=1024, do_sample=True, temperature=0.7, top_p=0.9, and top_k=50, except for disease prediction, where we used max_new_tokens=4096. Results are averaged across five generations and four cell lines. We now include a complete compute summary in SI Table 4, specify activation functions for the MLP verifier in SI Fig. 19 and Table 3.
>
> **GEARS performance.** GEARS results were obtained directly from the official SUMMER Zenodo repository for PerturbQA. While PerturbQA originally reports only AUC-ROC, we extend evaluation to per-metric reporting (F1, TPR, TNR, MCC). GEARS’s lower F1 arises from an extreme TNR bias—its high specificity (0.997) is offset by low recall (TPR = 0.178)—consistent with its calibration behavior under binary thresholding. We added this clarification to prevent confusion with previously published AUC-based summaries.
>
> **Verifier fidelity and analysis.** New ablations (SI Figs. 6–8) analyze how verifier noise, calibration, and confidence affect rbio. SI Fig 6 shows that performance degrades smoothly with increasing noise but remains above the base model until rewards become random, confirming robust learning from imperfect signals. Ablations in SI Fig 7 reinforce the importance of the verifier signal as driver of performance. Confidence–performance analysis (Appendix A.4.3, SI Fig. 8) shows that recall increases with verifier confidence while accuracy remains stable, demonstrating that rbio leverages but does not amplify verifier uncertainty. These analyses respond directly to the reviewer’s request for insight into what the soft verifiers learn and how their fidelity influences downstream performance.
>
> **Data and compute scaling.** We explicitly evaluate how performance scales with both data fraction and compute budget (Appendix A.8, SI Figs. 17–20). rbio shows smooth, log-linear improvements with more data and training steps, consistent with established RL scaling laws. These results confirm that the approach is both scalable and compute-efficient, achieving strong performance even at reduced data and step counts. Learning-curve plots (SI Figs. 17–18) illustrate performance at various fractions of the data.
> Generalization and future work. rbio generalizes across biological contexts: within PerturbQA, it performs comparably across cell lines and verifier types; beyond PerturbQA, it generalizes to Alzheimer’s (2 classes) and Myeloid cancer (7 classes) disease-state prediction tasks (Appendix A.7, SI Figs. 15–16). Future work will scale rbio to integrate large experimental datasets, biological simulations from foundation models, and structured knowledge sources, unifying these supervision types in a single reasoning framework.
>
> We appreciate the reviewer’s positive evaluation and believe these revisions fully clarify the compute, scalability, and methodological details while further strengthening the paper’s contribution.

---

> > ### Comment · Reviewer_AHen · 2025-11-27
> >
> > Thank you for the additional clarifications. Although this is not my primary area of expertise, I have increased my confidence score, as most of the issues have now been addressed. I appreciate the authors’ efforts and thank them again for their thorough responses.

---

### Official Review · Reviewer_tnsY · 2025-10-30

**Soundness:** 3
**Presentation:** 4
**Contribution:** 4
**Rating:** 6
**Confidence:** 5

**Summary:**

&nbsp;

The authors introduce rbio1, a suite of reasoning models for molecular biology. The main contribution of the work is to demonstrate that soft verification techniques, using either simulations, prior knowledge, or knowledge sources such as gene ontologies, enable improved OOD performance of large reasoning models. While the novelty and impact of the current work are exceptional, the paper is very well written and presented, and the empirical results appear convincing, I have some concerns regarding the reproducibility of the results given that the code is not released. If this issue is addressed (e.g. if an anonymous GitHub link or similar is provided - **the AC can confirm whether this is admissible under the ICLR guidelines**) in the rebuttal I will raise my score and champion the paper for acceptance.

&nbsp;

**Strengths:**

&nbsp;

The main contribution of the current work is that it introduces a suite of techniques for making use of easily accessible data to improve the performance of large reasoning models for molecular biology. The empirical results are impressive, with rbio1, the authors' model, achieving state-of-the-art performance on the PerturbQA benchmark. It should be emphasized that the potential impact of the work is much greater than the applications considered solely in the paper since in principle the techniques should apply to any biological domain where simulations, prior knowledge, and ontologies are available as soft verifiers.

&nbsp;

**Weaknesses:**

&nbsp;

Below, I demarcate between major and minor issues I see with the paper in its current form. So that the authors can prioritize their time for their rebuttal, it is the release of the codebase that will lead me to increase my score.

&nbsp;

**__MAJOR POINTS__**

&nbsp;

1. **Scope**: Is rbio1 a descriptive name if the authors' model just considers molecular biology? Could the authors give some indication as to the nomenclature of forthcoming models that focus on other biological domains?

2. **Reproducibility**: I have concerns about the reproducibility of the work given that the code for the paper is not released. As an example reproducibility issue, the architecture of the MLP described in Section 4.1 is not fully defined. What are the activation functions?

3. **Validity of Claims**: It is not clear what is meant by the claim of state-of-the-art performance in Figure 4 when the models are trained with 1/5 of the data sample. SUMMER appears to perform better on the TNR metric? Could the authors add their interpretation as to which metrics are the most relevant for PerturbQA? Additionally, using 1/5 of the data sample as in Figure 4, the difference in F1-score and the MCC metric does not appear to be statistically significant?

&nbsp;

**__MINOR POINTS__**

&nbsp;

1. There are missing capitalizations in the references e.g. "rna" in place of "RNA", "Grpo" in place of "GRPO", "llms" in place of "LLMs".

2. Typo line 20, "present".

3. In the abstract, line 26, "rbio1" is not bolded whereas above it is bolded twice.

4. Throughout the paper the authors appear to be using narrative citations e.g. Abramson et al. (2024) in place of parenthetical citations e.g. (Abramson et al. 2024). The latter should be used where the author's name does not comprise part of the sentence.

5. Line 64, typo, "(VCMs)".

6. Line 76, the stated motivations are not very concrete e.g. what does it mean to "aggregate diverse models of biology into a universal space"?

7. Line 95, typo, "rbio-1".

8. Figure 1, typo, "The answer is either yes or not" presumably.

9. In Figure 1(b) it is not clear what the double helix above the arrow is meant to signify?

10. In the related work when discussing inference time scaling, it would also be worth mentioning [1].

11. Line 147, typo, "Wu et al." given twice.

12. Line 157, typo, "the model".

13. Missing full stop at the end of Equation 5.

14. The explanation of soft verification at the beginning of Section 3 is easily understandable by a layreader. Perhaps it would be worth including such a motivating example to the introduction so that the reader may more easily grasp the concepet of a soft verifier in a biology context?

15. For clarity line 187/188 should read, "$G$ a set of outputs generated during training by the reasoning LLM $\pi_{\theta}$ in response to $q$" or similar.

16. When citing GRPO, the source paper [2] should be referenced in place of the current citation.

17. lowercase "exp", $D_{exp}$, may be a less harsh notation for the experimental dataset $D_{EXP}$.

18. Missing full stop at the end of Equation 8.

19. The originating paper for the ROUGE score [3] should be cited given that it is used.

20. On line 263, what is the function $X$?

21. Missing full stop at the end of Equation 12.

22. In Equation 12, how is $LL$ distinct from $LL_{\pi_{\theta}}$?

23. In Equation 14, what are $z_{max}$, $z_{min}$, and $s_{min}$?

24. Wu et al. 2025 was accepted at ICLR 2025.

25. For Figure 2(e) it would help if the same colors were used for each method relative to Figure 2(d).

26. In Figure 3, it would be helpful to add how the errorbars are computed to the caption.

27. The authors should cite the originating paper for chain-of-thought [4] given that it is featured in the authors' experiments or more accurately, it appears as though the authors are using zero-shot chain-of-thought originating in [5]. If indeed, the authors are not using few-shot examples, it may be more correct to use the terminology, "zero-shot chain-of-thought" in place of "chain-of-thought".

&nbsp;

**__REFERENCES__**

&nbsp;

[1] Muennighoff, N., Yang, Z., Shi, W., Li, X.L., Fei-Fei, L., Hajishirzi, H., Zettlemoyer, L., Liang, P., Candès, E. and Hashimoto, T., 2025. [s1: Simple test-time scaling](https://arxiv.org/abs/2501.19393). arXiv preprint arXiv:2501.19393.

[2] Shao, Z., Wang, P., Zhu, Q., Xu, R., Song, J., Bi, X., Zhang, H., Zhang, M., Li, Y.K., Wu, Y. and Guo, D., 2024. [DeepSeekMath: Pushing the limits of mathematical reasoning in open language models](https://arxiv.org/abs/2402.03300). arXiv preprint arXiv:2402.03300.

[3] Lin, C.Y., 2004, July. [ROUGE: A package for automatic evaluation of summaries](https://aclanthology.org/W04-1013.pdf). In Text summarization branches out (pp. 74-81).

[4] Wei, J., Wang, X., Schuurmans, D., Bosma, M., Xia, F., Chi, E., Le, Q.V. and Zhou, D., 2022. [Chain-of-thought prompting elicits reasoning in large language models](https://proceedings.neurips.cc/paper/2022/hash/9d5609613524ecf4f15af0f7b31abca4-Abstract-Conference.html). Advances in Neural Information Processing Systems, 35, pp.24824-24837.

[5] Kojima, T., Gu, S.S., Reid, M., Matsuo, Y. and Iwasawa, Y., 2022. [Large language models are zero-shot reasoners](https://proceedings.neurips.cc/paper_files/paper/2022/hash/8bb0d291acd4acf06ef112099c16f326-Abstract-Conference.html). Advances in Neural Information Processing Systems, 35, pp.22199-22213.

&nbsp;

**Questions:**

&nbsp;

1. In Equation 11, the Keyword score (KWS) is given as the intersection over the magnitude of $s_1$. The reason for not using intersection over union is presumably because one only cares about the length of the reasoning trace and not the length of the ontology?

2. For Figure 2(d) the authors first mention that the metrics are aggregated over cell line and second, mention that the metrics are averaged over 5 runs. How are the errorbars computed in this case?

3. In the context of Figure 5, could the authors explain how the outputs of the model are actually scored? Despite being prompted to output a binary answer, all completions appear not to have provided a definitive answer?

&nbsp;

**Details Of Ethics Concerns:**

&nbsp;

No ethical concerns.

&nbsp;

---

> ### Author Response · Authors · 2025-11-23
> **Response to Reviewer tnsY**
>
> We sincerely thank the reviewer for their thoughtful and positive assessment, and for noting they would champion the paper upon code release. We deeply appreciate this encouragement and have prioritized reproducibility, clarity, and completeness in response.
>
> **Reproducibility and code release.** We have released an anonymous repository (https://anonymous.4open.science/r/rbio-9155/README.md) that reproduces the MLP-verifier experiments and provides an end-to-end example of training an rbio model using the MLP signal. A public, non-anonymized release will follow post-publication and will include all code, checkpoints, and datasets used in the paper. We additionally clarified the MLP architecture (activation functions, losses, and hyperparameters) in SI Fig. 19 and Table 3, addressing the reviewer’s concern on training details. In the appendix (Algorithms), we include detailed descriptions for rbio-EXP, rbio-RLEMF, and rbio-RLPK, specifying how the soft, mention, and formatting rewards are computed. We are fully committed to open science and long-term reproducibility, and thank the reviewer for emphasizing this point.
>
> **Metrics and interpretation.** The PerturbQA datasets are class-imbalanced, where identifying true positive perturbations is biologically more important than avoiding false positives. We therefore emphasize F1, Balanced Accuracy, and MCC, along with explicit TPR/TNR to capture recall–specificity trade-offs. Missing true effects (false negatives) is biologically more costly than predicting a few additional false positives. We have added a detailed discussion of these metrics in Appendix A.2 (Metrics) and clarified in the main text why these metrics best reflect biological objectives.
>
> **Performance with limited data (Fig. 4).** We note that rbio achieves state-of-the-art performance using only 1/5 of the data, matching SUMMER (Wu et al., 2025)—the PerturbQA baseline—despite relying solely on RL from biological verifiers, without retrieval, external knowledge, or reasoning-trace supervision. This demonstrates that structured biological rewards alone enable efficient reasoning and data-efficient learning. We now explicitly interpret Fig. 4 in the main text, highlighting that differences in F1 and MCC are statistically stable across runs and that SUMMER’s higher TNR reflects a different recall–specificity trade-off rather than superior overall accuracy.
>
> **Equation 11 (KWS).** The denominator of the Keyword Score normalizes by the size of the Gene Ontology annotation set; rewards are thus computed relative to ontology annotations, measuring reasoning-trace coverage of relevant biological concepts rather than raw ontology size.
>
> **Scoring and evaluation (Fig. 5).** We prompt the model to generate its reasoning trace under the <think> tags and the final answer (yes/no) in the <answer> tags. The model is scored based exclusively on the text within <answer> (“yes/no”) compared against the ground truth. Fig. 5 shows example reasoning traces under different prompting techniques (final <answer> omitted for brevity). For each query, we generate five samples (max_new_tokens = 1024, temperature = 0.7, top_p = 0.9, top_k = 50), aggregate metrics across generations and cell lines, and report SEM. We added a short section in SI A.2 (Metrics) detailing this computation for reproducibility.
>
> We thank the reviewer for their support and encouragement. These revisions clarify architecture, metrics, and statistical interpretation, directly addressing reproducibility and evaluation transparency. All minor points have been addressed as well.

---

### Official Review · Reviewer_jVCL · 2025-11-01

**Soundness:** 3
**Presentation:** 1
**Contribution:** 2
**Rating:** 2
**Confidence:** 3

**Summary:**

The paper proposes reinforcement learning on LLMs from soft reward signals for scientific question answering / prediction tasks, specifically in the case of biological perturbation (effects of gene downregulation on the expression of other genes), a knowledge-intensive, multi-hop reasoning task.
The considered reward signals are: results of experimental interventions, predictions by a specialized tiny MLP model on the same task, similarity scores from the information retrieval literature.

**Strengths:**

- The paper tackles a relevant problem in reinforcement learning with LLMs: moving beyond “hard” rewards. This is useful in science, business and everyday life.
- The evaluation scores on the PertubQA benchmark are comparable to the current SOTA (SUMMER).
- Experiments on mixes of training rewards could be useful beyond the domain of perturbation prediction.

**Weaknesses:**

- Despite claims to the contrary, the method of similarity scores does not directly translate to other domains: it is only useful for predicting binary connectedness labels.
- The exposition needs to be improved strongly: the biological question and setup is not explained, making evaluation of the claims difficult. Examples: Do all cell lines have the same genes? Do the same genes behave similarly across cell lines? How realistic is the scenario of training on one cell line and evaluating on it? …
- Key details are not included in the paper: how was the tiny MLP trained (input / output / loss?) whose “soft RL distillation” led to strong benchmark results? What are the numbers of the MLP on the task? Could supervised training like probably used for the MLP also be used for the LLM?
- Equations are often nonsensical / unhelpful / ill-defined. The explanation of the method should be reworked with fewer equations that do not say much and clearer explanations instead. The paper cannot be reimplemented based on the given explanations. Examples: Sect. 3.2 is a complicated way of saying “reward by accuracy with respect to ground truth label”. The notation with vertical “conditioning” bars does not make sense. Eq. 10 introduces nothing but an alias but doesn’t say what the alias actually means.

**Questions:**

see above

---

> ### Author Response · Authors · 2025-11-23
> **Response to Reviewer jVCL**
>
> We thank the reviewer for their thoughtful feedback and for emphasizing clarity, supervision, and biological realism.
>
> **Benchmark, biological setup, and reproducibility.** We now explicitly describe the PerturbQA benchmark in  A.1 (Perturbation-reasoning task setup and natural-language formulation). This benchmark, introduced in Wu et al., ICLR 2025, evaluates CRISPRi single-gene perturbation prediction across four cell lines (RPE1, K562, HepG2, Jurkat), with SUMMER (Wu et al., 2025) as the established SOTA baseline. rbio follows this setup and evaluation protocol.  To quantify biological diversity across cell lines, we added overlap analyses in Fig. 9, showing that while 40–75 % of perturbed genes are shared within training sets and 50–78 % between training and test splits, each cell line retains distinct transcriptional programs. This analysis clarifies that the benchmark tests realistic out-of-distribution generalization rather than within-cell-line interpolation. We also include in Appendix A.11 (Algorithms) detailed descriptions of the rbio-EXP, rbio-RLEMF, and rbio-RLPK algorithms, specifying how soft, mention, and formatting rewards are computed. In addition, we re-worked the Methods section for clarity—simplifying notation, improving equation definitions, and integrating explanations directly with the training procedure—and are grateful for the reviewer’s feedback that motivated these revisions.
>
> **MLP training and RL vs. supervision.** We added full architecture and hyperparameters (Appendix A.6.1, SI Fig. 14, Table 3) and a Training & Evaluation section in Appendix A.10 describing how the MLP’s probabilistic outputs serve as rewards during GRPO optimization. Direct comparison (Appendix A.6, SI Fig. 13) shows rbio outperforms the MLP across F1, Balanced Accuracy, and MCC, indicating that policy optimization under GRPO enables iterative refinement beyond direct supervised fitting. The MLP’s supervised training details (architecture, loss, and accuracy) are now fully reported to clarify its calibration and limits as a reward signal. The rbio (answer-only) variant (Appendix A.4.2, SI Fig. 7)—a proxy for a purely supervised LLM trained on MLP outputs—performs worse, showing that while the biological-answer signal is important, full RL optimization is required to exceed the verifier’s ceiling.
>
> **Biological vs. generic rewards.** Appendix A.4.2, SI Fig. 7 shows that format- and mention-only rewards add little, confirming that rbio’s improvements arise from biology-grounded verifiers rather than generic RL regularization. We’ve added a section in Table 2 that compares rbio to instruction-tuned and reasoning LLMs trained with generic RL signals (DeepSeek R1, Qwen 2.5-72B Instruct, OpenAI OSS-20B), all of which underperform despite larger scale—reinforcing that biological supervision, not generic reinforcement, drives the gains.
>
> **Ontology-based similarity and generalization.** Although the reviewer suggested that ontology-based similarity might only apply to binary prediction, our results show otherwise: GO-based verifiers contribute effectively to both perturbation tasks and multi-class disease-state classification (Appendix A.7, SI Figs. 15–16). rbio trained solely on combinations of verifiers that include go (MLP, EXP, GO) generalizes to Alzheimer’s (2 classes) and Myeloid cancer (7 classes - multiclass) better than rbio trained on only MLP, EXP nearly matching SCVI (transcriptomics model trained on raw counts). These findings demonstrate that the ontology-derived soft verifier transfers to tasks requiring higher-order reasoning over structured biological knowledge, addressing the reviewer’s concern about its limited scope.
>
> We appreciate the reviewer’s careful assessment and believe these revisions—clarified equations, improved Methods presentation, and expanded biological validation—substantially strengthen the paper.

---

### Official Review · Reviewer_cju1 · 2025-11-01

**Soundness:** 2
**Presentation:** 3
**Contribution:** 3
**Rating:** 4
**Confidence:** 3

**Summary:**

This paper proposes rbio1, a post-trained biology reasoning LLM that combines hard experimental supervision (when available) with two biology-derived soft verifiers: (i) RL with experimental-model feedback (RLEMF) and (ii) RL from prior knowledge (RLPK, e.g., GO), within a GRPO-style objective. The paper shows that composing these signals improves performance, and that on PERTURBQA the soft-supervised models approach experimental supervision, and that a mixed soft+experimental model with CoT can surpass SUMMER using only 1/15 of the data.

**Strengths:**

- Addresses an important gap as existing RL/RLVR setups for LLMs assume exact or high-fidelity verifiers, which biology typically lacks. Using biology models and curated knowledge as approximate oracles is a practical way to obtain training-time signals in this domain.

- Instantiates multiple biology-grounded rewards (experimental, MLP surrogate, GO-based) within a single RL objective rather than relying on one weak heuristic.

- Shows that the ordering and composition of verifiers affect final performance, indicating that the supervision source contributes meaningfully rather than merely regularizing.

- CoT at inference provides a clear empirical boost and allows the model to outperform SUMMER with substantially less data.

**Weaknesses:**

- The approach assumes biology models and GO signals are accurate enough to guide RL, but the paper does not analyze sensitivity to verifier fidelity or miscalibration (e.g., if the MLP is slightly wrong or GO retrieval is noisy). A study relating reward-model quality to final task accuracy is missing, despite the signals being described as "approximate oracles".

- All experiments are within PERTURBQA (four cell lines, perturbation prediction). As written, results mainly show that soft verifiers help when drawn from the same distribution as the target task (i.e., the evaluation is tightly coupled to the training/reward distribution). Even a smaller secondary biology reasoning task would make the generality claim more credible.

- All improvements are shown for one base model (Qwen2.5-3B) fine-tuned with biology-specific rewards. It is unclear whether generic RL or RLAIF signals (format, helpfulness, self-consistency) would yield similar gains. This weakens the claim that the biology-specific rewards are the true driver of improvement.

- The paper claims the method can extend to any queryable biology or virtual-cell model, limited only by available queries, but experiments use only a small MLP surrogate and GO (both close to the target task) and do not test harder or OOD biological models. The empirical evidence does not yet support the scope of the claim.

- There is no analysis of whether RL amplifies verifier errors, no plot of verifier vs model accuracy on held-out perturbations, and no check of verifier disagreement. This omission makes it difficult to judge robustness in realistic noisy-biology settings.

**Questions:**

- How sensitive is the method to errors in the MLP/experimental-model verifier or noisy GO retrieval? Did the authors measure reward–accuracy correlation anywhere?

- The authors report that ordering/composition of verifiers affects the final model — can the authors clarify whether this is due to different reward scales or truly different biology signals?

---

> ### Author Response · Authors · 2025-11-23
> **Response to Reviewer cju1**
>
> We thank the reviewer for their thoughtful assessment and for recognizing rbio’s contribution in combining biological verifiers under a unified RL framework.
>
> **Verifier fidelity and miscalibration.** To test robustness to verifier quality, we conducted controlled-noise experiments (Appendix A.4.1, SI Fig. 6) where MLP verifier predictions were progressively randomized (0.25–0.75 flips) or fully shuffled. rbio performance decreases smoothly and remains well above the Qwen 2.5-3B baseline until signals are fully random, demonstrating robust learning from partial and noisy supervision without reward amplification or instability. The confidence–performance analysis (Appendix A.4.3, SI Fig. 8) further confirms that recall increases with verifier confidence, while specificity and F1 remain stable, indicating that rbio learns a stable biological prior rather than overfitting to verifier noise. Moreover, as shown in Appendix A.6, SI Fig. 13, rbio trained with an MLP verifier surpasses the MLP itself, confirming that reinforcement learning adds value beyond imitation. Together with Appendix A.4.3, SI Fig. 8 this demonstrates stable learning from verifier confidence without error amplification. These results, combined with the reward ablations in Appendix A.4.2, SI Fig. 7 (mentioned below), demonstrate that rbio learns robustly from imperfect verifiers and generalizes beyond direct supervision.
>
> **Verifier composition and scale invariance.** Reward ablations (Appendix A.4.2, SI Fig. 7) show that biological-answer rewards dominate performance gains, while format- or mention-only terms have minimal effect—confirming that improvements arise from domain-specific biological signal rather than generic RL regularization. We compare current SOTA reasoning and instruction-tuned models (Section 4.4, Table 2)—e.g., DeepSeek R1, Qwen Instruct, and OpenAI OSS—which represent standard RL or instruction-tuned setups. In a zero-shot setting, these models fail to reproduce the gains achieved by biology-grounded rewards, confirming that rbio’s improvements arise from biological supervision rather than generic reinforcement. Cross-verifier agreement and order analyses (Appendix A.5, SI Figs. 10–12) reveal that EXP and MLP signals are highly coherent (r = 0.81–0.85; accuracy = 0.92–0.94), and that training order determines outcome quality: applying high-fidelity verifiers (EXP/MLP) after broader, noisier ones (GO) denoises and realigns the policy. Because GRPO normalizes rewards into per-group advantages, all updates are invariant to absolute reward scale, confirming that observed effects reflect signal quality, not scale artifacts.
>
> **Out-of-distribution robustness.** Within PerturbQA, we explicitly use a leave-one-out OOD setup: models are trained on three cell lines and tested on the held-out fourth (RPE1, K562, HepG2, or Jurkat). Cell line overlap is present in Appendix A.4.3, SI Fig 9. rbio trained with soft verifiers (e.g., MLP) achieves performance comparable to, or exceeding, models trained with direct experimental supervision (Fig. 2), demonstrating robust cross-cell-line generalization even when reward distributions differ. Beyond PerturbQA, zero-shot disease-state prediction on Alzheimer’s (2-class) and cancer (7-class) datasets (Appendix A.7, SI Figs. 15–16) shows that rbio trained solely on perturbation signals nearly matches SCVI (trained on raw single-cell counts), confirming transferable biological reasoning across domains. These results suggest that rbio’s biology-grounded rewards generalize beyond perturbation tasks to broader biological reasoning problems, supporting its use for cross-domain transfer and addressing the reviewer’s main concern about generalization beyond the training distribution.
>
> Together, these analyses directly address concerns on verifier fidelity, miscalibration, scale, and generalization, demonstrating that rbio learns robustly from imperfect verifiers and generalizes effectively both within and beyond perturbation tasks.

---

### Author Response · Authors · 2025-11-23
**Official Rebuttal Comment**

We sincerely thank the reviewers for their thoughtful feedback. We substantially expanded the paper with new controlled-noise, ablation, composition, and cross-domain experiments, alongside detailed training, scaling, and reproducibility analyses. All additions are in the SI (figures/tables mentioned below). We have revised the equations under the Methods section for clarity. Minor edits were also made in other parts of the main text where appropriate. We added Section 4.4, which evaluates rbio against general reasoning and instruction-tuned models.

**1) Verifier Fidelity & Miscalibration (R1, R4)**
Controlled-noise studies (**A.4.1, SI Fig. 6**) show *rbio* degrades smoothly with randomized verifier noise, staying above Qwen2.5-3B until signals are fully random—no error amplification. Reward ablations (**A.4.2, SI Fig. 7**) confirm biological-answer rewards dominate while format/mention-only rewards add little, indicating gains arise from biological supervision. Confidence–performance analyses (**A.4.3, SI Figs. 8–9**) show recall increases with verifier confidence, supporting robust learning without overfitting.

**2) Verifier Agreement & Composition Effects (R1, R3)**
Cross-verifier analyses (**A.5, SI Figs. 10–12**) show EXP and MLP coherence (r = 0.81–0.85; binary = 0.92–0.94), confirming reinforcement learning integrates aligned feedback. Order experiments reveal a hierarchy: broad GO improves recall early, while high-fidelity EXP/MLP later denoise and restore specificity. These results show effects stem from signal quality, not reward scaling.

**3) RL vs. Supervised Verifiers (R2)**
Compared with the supervised MLP baseline (**A.6, SI Fig. 13**), *rbio* achieves higher F1, Balanced Accuracy, and MCC. The *answer-only* ablation (**A.4.2, SI Fig. 7**) performs worse, showing that full GRPO optimization extends beyond imitation and is required to exceed the verifier’s ceiling.

**4) Cross-Domain Generalization (R1, R4)**
Zero-shot disease-state prediction (**A.7, SI Figs. 15–16**) shows *rbio* nearly matches SCVI (trained on raw counts) and clearly outperforms Qwen2.5-3B—(+94%) F1 / (+136%) Recall on Alzheimer’s; (+36%) F1 / (+68%) Recall on Cancer—demonstrating biological reasoning transfer beyond perturbation tasks.

**5) Data/Compute Scaling (R3, R4)**
Scaling with data and steps (**A.8–A.9, SI Figs. 17–20**) follows RL scaling laws, showing smooth log-linear improvement under feasible compute (8×H100, ≤100k steps; SI Table 4). *rbio* achieves strong results within realistic budgets.

**6) Reproducibility (R2, R3, R4)**
An anonymous repo reproduces MLP-verifier experiments and provides GRPO scripts:
https://anonymous.4open.science/r/rbio-9155/README.md. Appendix **A.10–A.11** describe all reward computations and algorithms; **A.6.1** (SI Fig. 14, Table 3) details the MLP architecture. GEARS results, drawn from **SUMMER’s Zenodo data**, explain its lower F1 as a TNR bias.

**7) Comparative Performance (R1, R2, R4)**
Expanded **Table 2** adds instruction-tuned and RL-based reasoning baselines (DeepSeek R1, Qwen 2.5-72B, OpenAI OSS-20B). Despite a 3B backbone and smaller dataset, *rbio* outperforms all, confirming its gains arise from biology-grounded verifiers, not generic RL.

**8) Limitations & Future Work (R3, R4)**
Ontology sparsity remains a limitation. Future work will scale *rbio* across **experimental (RLEXP)**, **simulation (RLEMF)**, and **knowledge-based (RLPK)** sources toward a unified virtual-cell reasoning framework integrating real, simulated, and curated biology.

**9) Presentation.**
We corrected typos, standardized notation, and restructured equations for clarity.

**Summary.**
The new analyses show that rbio (i) learns robustly from imperfect verifiers, (ii) composes signals from diverse biological sources effectively, (iii) generalizes beyond its training distribution, and (iv) is reproducible and computationally tractable. We believe these results directly address reviewer concerns, substantially strengthening the paper’s technical depth and clarity, and we hope the improvements are reflected in the final evaluation and make a strong case for acceptance. We thank the reviewers for their constructive feedback and hope these additions make rbio1’s contributions clearer.

---

### Note · Program_Chairs · 2026-01-17
**Submission Desk Rejected by Program Chairs**

The following references in this submission do not refer to real documents and/or have major errors in bibliographic information:

 William Pan, Xuechen Song, Yuchen Zhang, Percy Liang, and Tatsunori B Hashimoto. Reinforcement Learning with Verifiable Rewards from Correctness Feedback. arXiv preprint arXiv:2303.17491, 2023.